# Beyond Entropy: Region Confidence Proxy for Wild Test-Time Adaptation

Zixuan Hu [1 2]   Yichun Hu [1]   Xiaotong Li [1]   Shixiang Tang [3]   Ling-Yu Duan [† 1 2]

## Abstract

Wild Test-Time Adaptation (WTTA) is proposed to adapt a source model to unseen domains under extreme data scarcity and multiple shifts. Previous approaches mainly focused on sample selection strategies, while overlooking the fundamental problem on underlying optimization. Initially, we critically analyze the widely-adopted entropy minimization framework in WTTA and uncover its significant limitations in noisy optimization dynamics that substantially hinder adaptation efficiency. Through our analysis, we identify *region confidence* as a superior alternative to traditional entropy, however, its direct optimization remains computationally prohibitive for real-time applications. In this paper, we introduce a novel region-integrated method **ReCAP** that bypasses the lengthy process. Specifically, we propose a probabilistic region modeling scheme that flexibly captures semantic changes in embedding space. Subsequently, we develop a *finite-to-infinite* asymptotic approximation that transforms the intractable region confidence into a tractable and upper-bounded proxy. These innovations significantly unlock the overlooked potential dynamics in local region in a concise solution. Our extensive experiments demonstrate the consistent superiority of ReCAP over existing methods across various datasets and wild scenarios. The source code will be available at https://github.com/hzcar/ReCAP.

## 1. Introduction

Deep neural networks have exhibited remarkable success across various visual tasks (Girshick, 2015; He et al., 2016). However, their performance is often compromised by the

[†]Corresponding author [1]School of Computer Science, Peking University, Beijing, China [2]Peng Cheng Laboratory, Shenzhen, China [3]The Chinese University of Hong Kong, Hongkong, China. Correspondence to: Ling-Yu Duan <lingyu@pku.edu.cn>.

*Proceedings of the 42nd International Conference on Machine Learning*, Vancouver, Canada. PMLR 267, 2025. Copyright 2025 by the author(s).

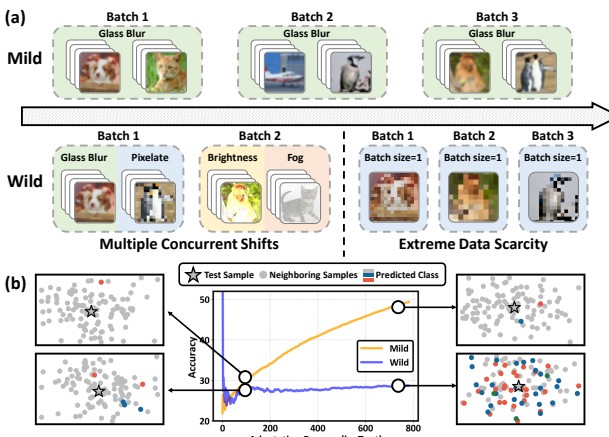

*Figure 1.* (a) Illustration of Mild (Wang et al., 2020) and Wild (Niu et al., 2023) TTA settings. (b) Comparison of the adaptation process between mild and wild scenes on the Zoom domain of ImageNet-C dataset (Hendrycks & Dietterich, 2019). Different colors of points represent different predicted classes of samples in the local region. The results highlight that entropy minimization in the wild scenario causes significant local prediction instability.

distribution shifts between training and testing data (Ben-David et al., 2010; Koh et al., 2021; Hu et al., 2024). To tackle this issue, Test-Time Adaptation (TTA) (Iwasawa & Matsuo, 2021; Alfarra et al., 2024; Liang et al., 2024) has emerged as a critical paradigm, enabling source models to adapt to target distributions through online updates. Its dominant approach involves optimizing the model to minimize prediction entropy, thereby enhancing the model's global confidence in the target domain.

While TTA methods (Zhou & Levine, 2021; Wang et al., 2022) have achieved promising results under mild conditions, they show significant performance drops in wild scenarios involving extreme data scarcity and multiple concurrent shifts (Niu et al., 2023), as shown in Fig. 1(a). To enable effective adaptation under these wild settings, recent works focus on developing selection criteria to leverage reliable samples only for entropy minimization. For example, SAR (Niu et al., 2023) excluded samples with high entropy and, DeYO (Lee et al., 2024) filtered out samples with sensitive prediction changes under image transformation.

Orthogonal to sample selection, this paper delves into a fundamental yet overlooked challenge: the noisy optimization dynamics introduced by typical entropy minimization.

In wild scenes, we observe a notable instability where the semantically similar samples within the local scope demonstrate a hard-to-compromise prediction discrepancy in wild scenes, as shown in Fig. 1(b). Such inconsistency leads the underlying optimization dynamics for these samples to become essentially conflicting. When entropy minimization is solely based on the individual sample, this narrow attention inevitably amplifies noisy dynamics, undermining both local consistency and overall adaptation efficiency.

Building on the above observations, it is essential to minimize the bias in the optimization direction, as well as the variance of unstable local predictions. Therefore, we propose a novel TTA strategy to enhance *region confidence*, which reflects the model's prediction certainty and consistency across the local region, rather than solely relying on biased individual predictions. We take two key statistical measures into consideration of the objective design: the *bias term* that quantifies the global entropy and the *variance term* that captures the prediction divergence within the region. These two terms work together to rectify overall optimization dynamics and reduce prediction disparity, promoting consistent adaptation across the entire region.

Despite the advantages of region confidence, the uncertain region scope and highly complex computations make it impractical for real-time testing. To overcome this, we introduce a new training framework, "**Re**gion **C**onfidence **A**daptive **P**roxy (**ReCAP**)", which incorporates a probabilistic region modeling mechanism and a highly efficient region optimization proxy. Specifically, ReCAP introduces a probabilistic representation to describe local regions as multivariate gaussian distribution, identifying a suitable region in feature space. Building upon this foundation, we propose a finite-to-infinite optimization proxy. Initially, we conduct a quantitative analysis of region confidence statistics under finite distribution sampling. Subsequently, we develop an asymptotic approximation to convert the intractable *bias* and *variance* terms into a concise, upper-bounded proxy. These upper bounds seamlessly integrate the extensive optimization dynamics of infinite local samples in a straightforward manner. As a result, our method establishes an efficient proxy for optimizing region confidence, replacing entropy-based approaches to unlock significantly enhanced adaptation efficiency.

We evaluate the effectiveness and generalizability of our method through experiments on both ResNet (He et al., 2016) and ViT (Dosovitskiy et al., 2020), achieving state-of-the-art results on ImageNet-C (Hendrycks & Dietterich, 2019) with average gains of +2.0%, +1.1%, +1.9% on 15 corruption shifts in three wild scenarios.

**Contributions.** 1) We analyze and verify the limitation of widely adopted entropy minimization in introducing conflicting dynamics in WTTA scenarios. To address this, we propose a superior alternative as *region confidence*, a novel training objective that leverages local knowledge to mitigate noisy conflicts. 2) To ensure real-time processing capability, we propose **ReCAP**, a novel training framework that incorporates two key components: a probabilistic modeling mechanism to flexibly capture variations in local region, and a finite-to-infinite asymptotic analysis to provide an efficient proxy for optimizing the intractable terms. 3) We demonstrate that ReCAP significantly outperforms existing WTTA methods through extensive experiments. Notably, ReCAP can seamlessly integrate with the orthogonal sample selection approaches in negligible computational overhead, showcasing a comprehensive framework for WTTA.

## 2. Related Work

We revisit the TTA methods for further analysis and put other related areas into the Appendix E due to space limits.

Test-Time Adaptation aims to enhance the performance on out-of-distribution samples during inference. Depending on whether the training process is altered, TTA methods can be mainly divided into two groups: 1) Test-Time Training (TTT) (Sun et al., 2020; Bartler et al., 2022; Hakim et al., 2023; Liu et al., 2024) jointly optimizes the model on training data with both supervised and self-supervised losses, and then conducts self-supervised training at test time. 2) Fully Test-Time Adaptation (Fully TTA) (Boudiaf et al., 2022; Hong et al., 2023; Zhao et al., 2023; Press et al., 2024; Hu et al., 2025) refrains from altering the training process and focuses solely on adapting the model during testing. In this paper, we focus on Fully TTA, as it is more generally applicable than TTT, allowing adaptation of arbitrary pre-trained models without access to training data.

Due to the broad applicability of TTA (Shin et al., 2022; Lin et al., 2023; Gao et al., 2024; Karmanov et al., 2024; Wang et al., 2024), a variety of methods have been developed. For instance, some methods adjust the affine coefficients of Batch Normalization layers to adapt to the target domain (Schneider et al., 2020; Hu et al., 2021; Lim et al., 2022). Others refine the prediction to provide a more robust training process (Zhang et al., 2022; Chen et al., 2022). Since Tent (Wang et al., 2020) introduces entropy minimization to enhance model confidence and reduce error rates, numerous works follow this practice of entropy-based training. Building upon Tent, SAR (Niu et al., 2023) and DeYO (Lee et al., 2024) propose selection strategies for Wild TTA scenes, which exclude harmful samples to enhance the accuracy of adaptation directions. In contrast to these selection approaches, this paper introduces a novel strategy that replaces entropy-based training with a framework designed to encourage and integrate consistent optimization dynamics, significantly enhancing adaptation efficiency.

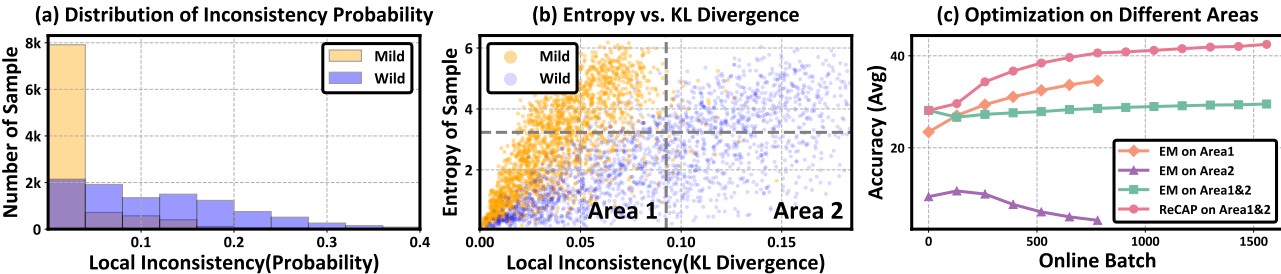

*Figure 2.* Local consistency during the entropy minimization process under mild and wild (imbalanced label shift) scenarios. Consistency is measured by prediction discrepancies between each sample and its 256 neighboring samples. (a) shows the probability of inconsistent predictions in neighbors. (b) records the entropy and average KL Divergence between prediction probabilities of samples and their neighbors. (c) investigates adaptation performance when using samples with varying levels of local consistency. All experiments are conducted on ImageNet-C of Gaussian noise with ResNet50. 'EM' denotes entropy minimization and 'ReCAP' denotes our method.

# 3. Local Inconsistency: A Barrier to Efficient Adaptation

## 3.1. Preliminaries for Wild Test-Time Adaptation

In Test-Time Adaptation (TTA), we have a model $F_\theta$ that has been pre-trained on the source domain $\mathcal{D}_\mathcal{S} = \{X_{train}, Y_{train}\}$ and need to evaluate it on the target domain $\mathcal{D}_\mathcal{T} = \{X_{test}, Y_{test}\}$. Here, $\theta$ denotes the model parameters, and $X, Y$ denote samples and labels with the distribution shift $P(X_{train}, Y_{train}) \neq P(X_{test}, Y_{test})$.

Unlike mild scenes in (Wang et al., 2020), Wild TTA tackles more complex environments involving extreme data scarcity and multiple concurrent shifts, including three practical scenarios: 1) Limited data stream, where batch size is restricted to 1. 2) Mixed testing domain, where target domain consists of $k$ different sub-domains: $\mathcal{D}_\mathcal{T}(X_{test}) = \sum_{i=1}^{k} \Pi_i \cdot \mathcal{D}_i$, with $\Pi_i$ being mixing coefficient. 3) Imbalanced label shift, where the test label distribution is imbalanced and shifts over time according to a function $Q_t(y)$.

To address these challenges, most existing methods (Niu et al., 2022; 2023; Lee et al., 2024) design various selection indicators to identify reliable samples and update $\theta$ through minimizing the entropy loss $\mathcal{L}_{ent}$:

$$\mathcal{L}_{ent}(x) = -p_\theta(x) \cdot \log p_\theta(x) = -\sum_{i=1}^{C} p_\theta(x)_i \log p_\theta(x)_i, \quad (1)$$

where $C$ is the number of classes and $p_\theta(x) = F_\theta(x) = (p_\theta(x)_1, \ldots, p_\theta(x)_C) \in \mathbb{R}^C$ is prediction probability on $x$.

## 3.2. Exploring Entropy Minimization via Local Consistency

Entropy minimization promotes the prediction probability to converge toward the dominant class, boosting confidence on unlabeled data. Its effectiveness heavily relies on the local consistency (nearby points share the similar prediction) to extend sample-wise confidence to a regional scale (Zhou et al., 2003; Wei et al., 2020). Intuitively, when predictions within a local space are consistent, optimization dynamics for individual points align with the overall direction, magnifying the local effects. Conversely, inconsistencies create

conflicting dynamics, introducing noise that hinders adaptation in the affected region. Such instability often stems from blurry decision boundaries and is prevalent in real-world deployments under domain shifts (Arani et al., 2022). Hence, it is crucial to evaluate the reliability of entropy minimization in preserving local consistency under wild scenes.

To assess local consistency, we record the prediction probabilities of test samples and their neighbors (sampled from the local region in feature space). From Fig. 2(a), the inconsistent probability converts from an unimodal distribution near zero to a fat-tailed distribution in wild scenes, reflecting a high risk of misaligned prediction within local space. Fig. 2(b) further reveals that while the entropy value is optimized to a similar level in both scenarios, the wild setting exhibits notably larger prediction discrepancies. These findings demonstrate that conventional entropy minimization can undermine local consistency.

Additionally, we evaluate the impact of using samples with varying levels of local consistency during adaptation, as shown in Fig. 2(c). Remarkably, entropy minimization using samples with low entropy and low consistency (Area 2) still carries performance collapse. Conversely, training with high consistency samples (Area 1) achieves comparable adaptation performance compared to joint training on Areas 1&2, showcasing superior adaptation efficiency. These results suggest that entropy minimization, even when combined with advanced selection, still hinders adaptation efficiency as it fails to ensure prediction consistency.

## 3.3. From Sample Confidence to Region Confidence

Building on the above findings, it is essential to address the bias between the optimization direction and regional objective, while also reducing the variance of inconsistent prediction probabilities within the local region. To this end, we introduce a novel objective called *region confidence* to replace vanilla entropy. This objective optimizes both region-wise confidence and stability simultaneously, thereby improving global optimization efficiency. The mathematical definition is as follows:

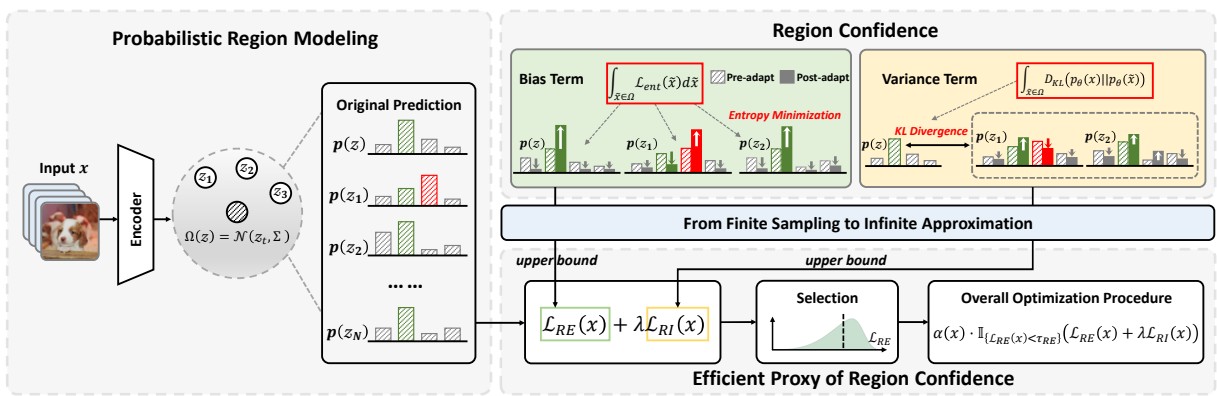

*Figure 3.* Overview of our ReCAP. ReCAP performs probabilistic modeling to determine local regions in the latent space (Sec. 4.1). We further derives two closed-form upper bounds for the intractable bias and variance terms via a finite-to-infinite asymptotic approximation, offering an efficient proxy for optimizing Region Confidence without the lengthy sampling process (Sec. 4.2 & 4.3).

**Definition 3.1. (Region Confidence)** Let us consider a local region $\Omega$ of a sample $x$. The region confidence of $x$ on $\Omega$ is defined as integrals of entropy loss on $\Omega$ (*Bias term*) plus the Kullback-Leibler divergence between prediction probabilities of $x$ and those of samples in $\Omega$ (*Variance term*):

$$\mathcal{L}_{RC}(x) = \underbrace{\int_{\Omega} \mathcal{L}_{\text{ent}}(\tilde{x})d\tilde{x}}_{\text{Bias term}} + \lambda \underbrace{\int_{\Omega} \mathcal{D}_{KL}(p_{\theta}(x) \| p_{\theta}(\tilde{x}))\, d\tilde{x}}_{\text{Variance term}},$$
$$(2)$$

where $\mathcal{D}_{KL}(p\|q) = \sum_{i=1}^{C} p_i \log \frac{p_i}{q_i}$ denotes the Kullback-Leibler divergence and $\lambda$ denotes a trade-off coefficient.

These two terms have distinct but complementary effects. The bias term integrates entropy loss over infinite samples in the local region, enabling an optimization that aligns with the overall training dynamics. The variance term penalizes large discrepancies, enhancing local consistency and reducing the dispersion of dynamics. By combining two terms, region confidence promotes consistent and confident predictions within the region, harnessing the potential dynamics embedded in the local space to boost adaptation efficiency.

## 4. Region Confidence Proxy

Despite the advantages of region confidence, there are still two considerable challenges to optimize it: 1) Uncertain region scope. The static regions fix the sample location, making it difficult to determine an appropriate local scope. 2) Heavy computational burden. Both terms in Eq. 2 are intractable, relying on extensive sampling and additional model forwards for measurement. To tackle these issues, we propose a new TTA method called "**Re**gion **C**onfidence **A**daptive **P**roxy (**ReCAP**)", which incorporates a probabilistic region modeling mechanism (Sec. 4.1) and an efficient proxy for optimizing region confidence (Sec. 4.2).

### 4.1. Probabilistic Region Modeling

Since different directions in feature space can imply potentials of valuable semantic changes (Bengio et al., 2013; Li

et al., 2023; Yu et al., 2024a), we model local regions in feature space to flexibly capture rich semantic information. To avoid class mixing, we further select the latent space extracted by the backbone, which ensures optimal class separability in the model. Hence, we explore region confidence within this latent space, and $p_{\theta}(x)$ in Eq. 2 can be replaced by the probability of its corresponding feature $z$:

$$p_{\theta}(z)_i = (\text{softmax}(Az + b))_i = \frac{e^{a_i \cdot z + b_i}}{\sum_{j=1}^{C} e^{a_j \cdot z + b_j}}, \quad (3)$$

where the subscript $(\cdot)_i$ denotes the $i$-th class. $A \in \mathbb{R}^{C \times d}$ and $b \in \mathbb{R}^C$ denote the linear and bias coefficients of the affine layer in the classifier, respectively.

Rather than treating the local region as a static range, we model it as a probabilistic representation following a multivariate Gaussian distribution. Specifically, the center of this distribution corresponds to each feature, while the variance, which defines the scope, is estimated using a small set of in-distribution data. The region is determined as follows:

$$\Omega(z_t) := \mathcal{N}(z_t, \tau \cdot \Sigma), \Sigma = \text{Diag}(\text{Var}_{\mathcal{D}_\mathcal{S}}(z)), \quad (4)$$

where $\Omega(z_t)$ is the local region of the t-th test batch $z_t$ and $\Sigma$ is the diagonal matrix of variance on a small set of source data, *e.g.*, 500 samples are enough for ImageNet-C dataset. $\tau$ is a hyper-parameter to control the scope.

### 4.2. Efficient Metric of Region Confidence

Based on the local region defined above, the computation of two terms depends on an infinite number of potential samples from the distribution and requires extensive sampling to capture the statistical information. However, the computational overhead and memory usage increase almost linearly with the number of sampling, making it impractical for real-time requirements in TTA testing. To address this issue, we develop finite-to-infinite approximation for the two terms, leading to a highly efficient implementation.

Taking the bias term as an example, we first consider the case of finite sampling:

**Lemma 4.1.** *(Bias Term under Finite Sampling) Given N features $z_1, \ldots, z_N$ drawn from the local region and their corresponding probabilities: $p_\theta(z_1), \ldots, p_\theta(z_N)$. The bias term on these features satisfies the following inequality:*

$$\sum_{i=1}^{N} \mathcal{L}_{ent}(p_\theta(z_i)) \leqslant - \sum_{i=1}^{C} \frac{\sum_{k=1}^{N} p_\theta(z_k)_i}{N} \cdot \left( \sum_{j=1}^{N} \log p_\theta(z_j)_i \right), \tag{5}$$

*where $p_\theta(z_i)$ is defined in Eq. 3.*

In the following, we consider the case where $N$ grows to infinity and find that $\frac{\sum_{k=1}^{N} p_\theta(z_k)_i}{N}$ in Eq. 10 gradually converges to the expectation of the prediction in the local region. For the remaining summation term $\sum_{j=1}^{N} \log p_\theta(z_j)_i$, it actually corresponds to the negative log-likelihood and can be scaled using the following lemma:

**Lemma 4.2.** *(Upper Bound of Negative Log-Likelihood) Given a feature $z$ and its local region $\Omega$ which follows a Gaussian distribution $\mathcal{N}(\mu, \Sigma)$. The expected value of the logarithm of the predicted probability for the i-th class satisfies the following inequality:*

$$- \mathbb{E}_{z \sim \mathcal{N}(\mu, \Sigma)} \log p_\theta(z)_i$$
$$\leq \log \sum_{j=1}^{C} e^{(a_j - a_i) \cdot \mu + (b_j - b_i) + \frac{1}{2}(a_j - a_i)\Sigma(a_j - a_i)^\top}, \tag{6}$$

*where the superscript $(\cdot)^\top$ denotes the transpose operation.*

Through the above two lemmas, we can further derive a closed-form upper bound for the bias term via asymptotic approximation. Refer to the Appendix A for missing proofs and detailed explanations.

**Proposition 4.3.** *(Efficient Metric of Bias Term) Given a feature $z$ and its local region $\Omega$ which follows a Gaussian distribution $\mathcal{N}(\mu, \Sigma)$. The expectation of entropy loss over the entire distribution has a closed-form upper bound:*

$$\mathbb{E}_\Omega[\mathcal{L}_{ent}] = - \mathbb{E}_{\tilde{z} \sim \mathcal{N}(z, \Sigma)} \sum_{i=1}^{C} p_\theta(\tilde{z})_i \log p_\theta(\tilde{z})_i$$
$$\leqslant \sum_{j=1}^{C} \frac{e^{a_j \cdot z + b_j + \frac{1}{2} a_j \sum a_j^\top}}{\sum_{k=1}^{C} e^{a_k \cdot z + b_k + \frac{1}{2} a_k \sum a_k^\top}} \log \sum_{i=1}^{C} e^{(b_i - b_j)}$$
$$\cdot e^{(a_i - a_j) \cdot z + \frac{1}{2}(a_i - a_j)\Sigma(a_i - a_j)^\top} \triangleq \mathcal{L}_{RE}. \tag{7}$$

*where the upper bound $\mathcal{L}_{RE}$ is called **Regional Entropy**.*

Notably, Proposition 4.3 offers an easy-to-compute metric without any additional sampling or model forward passes. Instead of minimizing the exact bias term in Eq. 2, we can optimize its upper bound to implicitly enhance overall sample confidence within the region with minimal cost.

Meanwhile, we apply a similar mathematical framework to derive a closed-form upper bound for the variance term.

**Proposition 4.4.** *(Efficient Metric of Variance Term) Given a feature $z$ and its local region $\Omega$ which follows a Gaussian*

*distribution $\mathcal{N}(\mu, \Sigma)$. The expectation of Kullback-Leibler divergence between the output probability over this distribution and the probability at its center has a upper bound:*

$$E_{\tilde{z} \sim \mathcal{N}(z, \Sigma)} KL(p_\theta(z) \| p_\theta(\tilde{z}))$$
$$\leq \sum_{j=1}^{C} \frac{e^{a_j \cdot z + b_j}}{\sum_{k=1}^{C} e^{a_k \cdot z + b_k}} \cdot \log \sum_{i=1}^{C} \frac{e^{a_i \cdot z + b_i}}{\sum_{k=1}^{C} e^{a_k \cdot z + b_k}} \tag{8}$$
$$\cdot e^{\frac{1}{2}(a_i - a_j) \sum (a_i - a_j)^\top} \triangleq \mathcal{L}_{RI},$$

*where the upper bound $\mathcal{L}_{RI}$ is called **Regional Instability**.*

This proposition also provides a theoretical result that captures local information without the need for sampling. By combining two upper bounds, we ultimately present an efficient proxy of region confidence in Eq.2.

### 4.3. Overall Procedure of ReCAP

Following common practices (Niu et al., 2022; Lee et al., 2024), the loss function requires filtering and weighting based on reliability. Unlike traditional entropy-based criteria, our analysis in Sec. 3.2 shows that regional confidence can also serve as a measure of reliability from an orthogonal perspective. Building on this insight, we leverage *Regional Entropy $\mathcal{L}_{RE}$* to identify reliable samples and enhance their optimization dynamics during adaptation. Formally, the overall procedure is as follows:

$$\min_\theta \alpha(x) \cdot \mathbb{I}_{\{\mathcal{L}_{RE}(x) < \tau_{RE}\}}(\mathcal{L}_{RE}(x) + \lambda \mathcal{L}_{RI}(x)),$$
$$\text{where } \alpha(x) \triangleq \frac{1}{exp(\mathcal{L}_{RE}(x) - \mathcal{L}_0)}, \tag{9}$$

where $\alpha(x)$ and $\mathbb{I}_{\{\mathcal{L}_{RE}(x) < \tau_{RE}\}}$ denotes the weighting and selection term. $\mathcal{L}_0$, $\tau_{RE}$ and $\lambda$ are hyper-parameters.

## 5. Experiments

### 5.1. Experimental Setup

For a fair comparison, we follow the identical network architectures, optimizer, and batch sizes as the Wild TTA benchmark proposed in (Niu et al., 2023).

**Datasets.** We conduct our experiments on three datasets to evaluate the robustness and generalization capability of our method under diverse distribution shifts: 1) ImageNet-C (Hendrycks & Dietterich, 2019), a large-scale dataset categorized into 15 common corruption types and 5 severity levels for each type. 2) ImageNet-R (Hendrycks et al., 2021) and 3) VisDA-2021 (Bashkirova et al., 2022), two datasets which encompass diverse domain shifts due to varying styles and textures (e.g., sketch, cartoon), compared to ImageNet-C to assess the efficacy for more challenging wild test scenarios in the Appendix B.

**Compared Methods.** We compare our ReCAP with the following state-of-the-art methods: DDA (Gao et al., 2022)

*Table 1.* Comparisons with state-of-the-art methods on ImageNet-C (severity level 5) under **Limited Batch Size = 1** regarding Accuracy (%). We report mean performance over 3 independent runs. The best results are in bold and the second-best are in underline.

| Model+Method | Noise | | | Blur | | | | Weather | | | | Digital | | | | Average |
|---|---|---|---|---|---|---|---|---|---|---|---|---|---|---|---|---|
| | Gauss. | Shot | Impul. | Defoc. | Glass | Motion | Zoom | Snow | Frost | Fog | Brit. | Contr. | Elastic | Pixel | JPEG | |
| ResNet50 | 18.0 | 19.8 | 17.9 | 19.8 | 11.4 | 21.4 | 24.9 | 40.4 | 47.3 | 33.6 | 69.3 | 36.3 | 18.6 | 28.4 | 52.3 | 30.6 |
| • MEMO | 18.5 | 20.5 | 18.4 | 17.1 | 12.6 | 21.8 | 26.9 | 40.4 | 47.0 | 34.4 | 69.5 | 36.5 | 19.2 | 32.1 | 53.3 | 31.2 |
| • DDA | 42.4 | 43.3 | 42.3 | 16.6 | 19.6 | 21.9 | 26.0 | 35.7 | 40.1 | 13.7 | 61.2 | 25.2 | 37.5 | 46.6 | 54.1 | 35.1 |
| • Tent | 2.5 | 2.9 | 2.5 | 13.5 | 3.6 | 18.6 | 17.6 | 15.3 | 23.0 | 1.4 | 70.4 | 42.2 | 6.2 | 49.2 | 53.8 | 21.5 |
| • EATA | 24.9 | 28.0 | 25.8 | 18.3 | 17.0 | 31.2 | 29.8 | 42.5 | 44.1 | 41.3 | 70.9 | 44.2 | 27.6 | 46.8 | 55.4 | 36.5 |
| • SAR | 25.5 | 28.0 | 24.9 | 18.7 | 16.3 | 28.6 | 31.4 | 46.2 | 44.9 | 33.4 | 72.8 | 44.3 | 15.3 | 47.1 | 56.1 | 35.6 |
| • DeYO | 41.2 | 44.3 | 42.5 | 22.4 | 24.7 | 41.8 | 21.9 | 54.8 | 51.6 | 21.9 | 73.1 | 53.2 | 48.5 | 59.8 | 59.6 | 44.1 |
| • ReCAP (Ours) | **42.5** | 44.4 | **42.9** | 19.4 | 25.0 | 42.2 | 44.0 | 49.7 | 52.4 | **57.5** | 72.9 | 53.6 | 29.5 | 60.4 | 60.0 | 46.4 |
| • ReCAP+SAR | 41.7 | 44.5 | 40.6 | 24.8 | 25.8 | **44.0** | **47.0** | 56.2 | 53.0 | 52.8 | 73.4 | 54.6 | 48.8 | **61.7** | 60.7 | 48.6 |
| • ReCAP+DeYO | **42.5** | **44.8** | 42.8 | **25.9** | **27.2** | 43.7 | 44.9 | 55.8 | 52.8 | 51.9 | **73.5** | **54.8** | **50.9** | 61.5 | **60.7** | **48.9** |
| Vitbase | 9.5 | 6.7 | 8.2 | 29.0 | 23.4 | 33.9 | 27.1 | 15.9 | 26.5 | 47.2 | 54.7 | 44.1 | 30.5 | 44.5 | 47.8 | 29.9 |
| • MEMO | 21.6 | 17.3 | 20.6 | 37.1 | 29.6 | 40.4 | 34.4 | 24.9 | 34.7 | 55.1 | 64.8 | 54.9 | 37.4 | 55.4 | 57.6 | 39.1 |
| • DDA | 41.3 | 41.1 | 40.7 | 24.4 | 27.2 | 30.6 | 26.9 | 18.3 | 27.5 | 34.6 | 50.1 | 32.4 | 42.3 | 52.2 | 52.6 | 36.1 |
| • Tent | 42.2 | 1.0 | 43.3 | 52.4 | 48.2 | 55.5 | 50.5 | 16.5 | 16.9 | 66.4 | 74.9 | 64.7 | 51.6 | 67.0 | 64.3 | 47.7 |
| • EATA | 30.1 | 24.6 | 34.2 | 44.3 | 39.6 | 48.4 | 42.4 | 38.1 | 46.0 | 60.7 | 65.8 | 61.2 | 46.7 | 57.8 | 59.5 | 46.6 |
| • SAR | 42.7 | 39.5 | 41.9 | 54.6 | 51.2 | 58.3 | 54.4 | 60.2 | 54.7 | 70.3 | 75.9 | 66.8 | 58.4 | 69.5 | 66.3 | 57.6 |
| • DeYO | 53.4 | 50.4 | 55.0 | 58.7 | 59.5 | 64.5 | 52.5 | 68.1 | 66.3 | 73.8 | 78.3 | 67.9 | 68.9 | 73.8 | 70.8 | 64.1 |
| • ReCAP (Ours) | 53.5 | **56.7** | **56.9** | 59.2 | 60.5 | 65.3 | 64.0 | 69.6 | 67.2 | 74.1 | 78.4 | 64.6 | 70.2 | 74.4 | 71.5 | 65.7 |
| • ReCAP+SAR | **55.3** | 56.2 | 56.5 | **60.0** | **61.2** | **66.5** | **65.2** | **69.8** | **68.0** | **74.5** | **78.6** | **68.4** | **71.0** | **74.9** | **71.6** | **66.5** |
| • ReCAP+DeYO | 54.4 | 55.3 | 55.5 | 59.8 | 61.1 | 65.2 | 64.9 | 69.3 | 67.9 | 73.9 | 78.4 | 67.1 | 70.7 | 74.4 | 71.0 | 65.9 |

performs diffusion-based adaptation to map the input image back to the source domain. MEMO (Zhang et al., 2022) minimizes marginal entropy across augmented variants of test samples. EATA, SAR, and DeYO are selection-based methods with distinct designs. EATA (Niu et al., 2022) combines entropy-based selection with a weighting mechanism. SAR (Niu et al., 2023) minimizes entropy using sharpness-aware optimization. DeYO (Lee et al., 2024) employs dual filtering based on disentangled factors.

*Table 2.* Comparisons with SOTA methods on ImageNet-C (severity level 5, 4) under **Mixed Testing Domain**.

| Model+Method | Level=5 | Level=4 | Average |
|---|---|---|---|
| ResNet50 | 30.6 | 42.7 | 36.7 |
| • MEMO | 31.2 | 43.0 | 37.1 |
| • DDA | 35.1 | 43.6 | 39.4 |
| • Tent | 13.4 | 20.6 | 17.0 |
| • EATA | 38.1 | 47.7 | 42.9 |
| • SAR | 38.3 | 48.6 | 43.5 |
| • DeYO | 38.6 | 50.2 | 44.4 |
| • ReCAP (Ours) | 41.5 | 51.2 | 46.4 |
| • ReCAP+SAR | 42.1 | 51.9 | 47.0 |
| • ReCAP+DeYO | **42.2** | **52.4** | **47.3** |
| VitBase | 29.9 | 42.9 | 36.4 |
| • MEMO | 39.1 | 51.3 | 45.2 |
| • DDA | 36.1 | 45.1 | 40.6 |
| • Tent | 16.5 | 64.3 | 40.4 |
| • EATA | 55.7 | 63.7 | 59.7 |
| • SAR | 57.1 | 64.9 | 61.0 |
| • DeYO | 59.4 | 66.8 | 63.1 |
| • ReCAP (Ours) | 59.4 | 67.1 | 63.3 |
| • ReCAP+SAR | **60.0** | **67.2** | **63.6** |
| • ReCAP+DeYO | 59.8 | 67.0 | 63.4 |

**Implementation Details.** Following the common setting in Wild TTA (Niu et al., 2023; Lee et al., 2024), we conduct experiments on ResNet50-GN (Wu & He, 2018) and ViTBase-LN (Dosovitskiy et al., 2020) obtained from `timm` (Wightman, 2019). For the optimizer, we use SGD, batch size of 64 (except for batch size=1), with a momentum of 0.9, and a learning rate of 0.00025/0.001 for ResNet/ViT. For our ReCAP, $\mathcal{L}_0$ and $\tau_{RE}$ in Eq. 9 is set to $0.7/1.0 \times \ln C$ and $0.8/1.0 \times \ln C$ ($C$ is the number of classes) for ResNet/ViT. The hyper-parameter $\tau$ in Eq. 4 is 1.2 and $\lambda$ in Eq. 9 is 0.5 by default. For trainable parameters, according to common practices (Wang et al., 2020), we adapt the affine parameters of normalization layers.

### 5.2. Evaluation Results

**Evaluation on Data Scarcity.** To evaluate the effectiveness of our method under severe data scarcity, we compare our ReCAP with prior approaches in challenging scenarios with limited data streams, *i.e.*, batch size = 1. As shown in Tab. 1, ReCAP significantly improves adaptation performance, emerging as the only method to consistently outperform the source model across all corruption types. Notably, ReCAP achieves superior results in 25 out of 30 cases, substantially surpassing the previous SOTA method by +2.3% and +1.6% on ResNet and ViT evaluations, respectively. These results underscore the robustness of our method in the face of data limitations and validate its effectiveness in enhancing adaptation efficiency for resource-constrained, low-data training.

**Evaluation on Multiple Concurrent Shifts.** To evaluate the ability of our ReCAP to handle complex distribution shifts, we compare various methods under mixed testing

*Table 3.* Comparisons with state-of-the-art methods on ImageNet-C (severity level 5) under **Imbalanced Label Shift**.

| Model+Method | Noise | | | Blur | | | | Weather | | | | Digital | | | | Average |
|---|---|---|---|---|---|---|---|---|---|---|---|---|---|---|---|---|
| | Gauss. | Shot | Impul. | Defoc. | Glass | Motion | Zoom | Snow | Frost | Fog | Brit. | Contr. | Elastic | Pixel | JPEG | |
| ResNet50 | 17.9 | 19.9 | 17.9 | 19.7 | 11.3 | 21.3 | 24.9 | 40.4 | 47.4 | 33.6 | 69.2 | 36.3 | 18.7 | 28.4 | 52.2 | 30.6 |
| • MEMO | 18.4 | 20.6 | 18.4 | 17.1 | 12.7 | 21.8 | 26.9 | 40.7 | 46.9 | 34.8 | 69.6 | 36.4 | 19.2 | 32.2 | 53.4 | 31.3 |
| • DDA | **42.5** | 43.4 | 42.3 | 16.5 | 19.4 | 21.9 | 26.1 | 35.8 | 40.2 | 13.7 | 61.3 | 25.2 | 37.3 | 46.9 | 54.3 | 35.1 |
| • Tent | 2.6 | 3.3 | 2.7 | 13.9 | 7.9 | 19.5 | 28.7 | 16.5 | 21.9 | 1.8 | 70.5 | 42.2 | 6.6 | 49.4 | 53.7 | 22.8 |
| • EATA | 27.2 | 28.5 | 28.4 | 15.1 | 16.7 | 24.6 | 25.5 | 32.5 | 32.2 | 40.0 | 66.5 | 33.2 | 24.1 | 42.2 | 38.6 | 31.7 |
| • SAR | 34.0 | 36.7 | 36.2 | 21.8 | 20.9 | 33.2 | 32.4 | 38.7 | 45.6 | 50.6 | 72.9 | 46.8 | 14.3 | 52.2 | 56.8 | 39.5 |
| • DeYO | 41.7 | 44.0 | 42.5 | 23.4 | 23.9 | 41.3 | 13.0 | 53.9 | 52.2 | 38.6 | 73.1 | 52.3 | 46.8 | 59.3 | 59.1 | 44.3 |
| • ReCAP (Ours) | 42.0 | 44.1 | 42.7 | 19.8 | 24.3 | 39.7 | 40.2 | 46.0 | 52.2 | 57.3 | 73.1 | 52.4 | 33.7 | 59.4 | 59.5 | 45.8 |
| • ReCAP+SAR | 42.2 | 44.4 | 42.7 | 24.1 | 24.3 | 41.6 | **43.8** | 51.6 | 52.4 | 56.8 | 73.1 | 52.9 | 38.9 | 60.2 | 59.4 | 47.2 |
| • ReCAP+DeYO | **42.5** | **44.5** | **43.3** | 25.8 | **26.7** | 43.3 | 39.1 | **54.2** | **53.2** | **59.3** | **73.4** | **53.8** | **49.2** | **61.4** | **60.3** | **48.7** |
| Vitbase | 9.4 | 6.7 | 8.3 | 29.1 | 23.4 | 34.0 | 27.0 | 15.8 | 26.3 | 47.4 | 54.7 | 43.9 | 30.5 | 44.5 | 47.6 | 29.9 |
| • MEMO | 21.6 | 17.4 | 20.6 | 37.1 | 29.6 | 40.6 | 34.4 | 25.0 | 34.8 | 55.2 | 65.0 | 54.9 | 37.4 | 55.5 | 57.7 | 39.1 |
| • DDA | 41.3 | 41.3 | 40.6 | 24.6 | 27.4 | 30.7 | 26.9 | 18.2 | 27.7 | 34.8 | 50.0 | 32.3 | 42.2 | 52.5 | 52.7 | 36.2 |
| • Tent | 32.7 | 1.4 | 34.6 | 54.4 | 52.3 | 58.2 | 52.2 | 7.7 | 12.0 | 69.3 | 76.1 | 66.1 | 56.7 | 69.4 | 66.4 | 47.3 |
| • EATA | 35.8 | 34.8 | 36.8 | 45.1 | 47.3 | 49.3 | 47.8 | 56.6 | 55.5 | 62.1 | 72.3 | 21.6 | 56.0 | 64.6 | 63.7 | 50.0 |
| • SAR | 48.2 | 48.7 | 49.0 | 55.4 | 54.5 | 59.2 | 54.3 | 55.8 | 54.5 | 70.0 | 76.9 | 66.1 | 62.2 | 70.2 | 66.5 | 59.4 |
| • DeYO | 53.0 | 34.4 | 48.8 | 57.6 | 58.5 | 63.3 | 35.4 | 67.4 | 66.0 | 73.0 | **77.7** | 66.6 | 68.1 | **73.1** | 69.8 | 60.8 |
| • ReCAP (Ours) | 53.1 | 38.5 | 49.6 | 57.3 | **59.0** | 63.8 | 60.7 | 67.8 | 66.3 | 72.9 | **77.7** | 66.8 | 68.2 | 73.0 | **70.0** | 63.0 |
| • ReCAP+SAR | **53.2** | 42.7 | 50.9 | **58.0** | 58.9 | 63.8 | 60.9 | 67.6 | 66.1 | **73.2** | 77.6 | 67.6 | **68.3** | **73.1** | 69.8 | 63.4 |
| • ReCAP+DeYO | 53.1 | **53.9** | **54.1** | 57.4 | 58.8 | 63.6 | 60.6 | 67.8 | **66.4** | 72.9 | **77.7** | 67.1 | **68.3** | **73.1** | 69.8 | **64.3** |

domain and imbalanced label shift. As shown in Tab. 2 & 3, Tent and MEMO struggle with multiple concurrent shifts, even performing worse than the no-adapt model. While selection methods like SAR and DeYO perform competitively in long-term adaptation scenarios, the limitations of entropy-based approaches still hinder their adaptation efficiency. In contrast, ReCAP achieves SOTA performance across nearly all corruption types. For label shifts, ReCAP showcases significant superiority over other methods, with an average gain of +1.5% and +2.2% in ResNet and ViT testing, respectively. These results validate the stable improvements offered by ReCAP under multiple concurrent corruptions.

**ReCAP can boost entropy-based methods.** To investigate the complementarity of our approach with prior entropy-based methods, we test its integration with the latest SOTA SAR and DeYO across all three wild scenarios. Through replacing entropy minimization with region confidence optimization proxy, the performance combined with our approach shows obvious gains in many downstream scenes.

*Table 4.* Efficiency comparison of various methods. We assess TTA approaches for processing 50,000 images in Gaussian corruption type, using a single Nvidia RTX 4090 GPU.

| Method | Forward | Backward | Other computation | Time |
|---|---|---|---|---|
| No-adapt | 50,000 | N/A | N/A | 84s |
| DDA | - | - | Diffusion model | 13,277s |
| Tent | 50,000 | 50,000 | N/A | 110s |
| EATA | 50,000 | 19,608 | regularizer | 118s |
| SAR | 66,418 | 30,488 | Model updates | 164s |
| DeYO | 82,843 | 24,714 | Data augmentation | 144s |
| ReCAP | 50,000 | 19,512 | Eq. 9 | 116s |

Specifically, our method consistently improves SAR with an average gain of **+11.0%**, **+3.1%**, and **+5.9%** across the three wild scenes. Similarly, DeYO achieves improvements of **+3.3%**, **+1.6%**, and **+4.0%** when integrated with our method. These significant gains validate the effectiveness of our proposed region confidence optimization strategy, which can seamlessly integrate with various methods to boost adaptation performance.

### 5.3. Running Time Comparison

We measure the running time required for the adaptation of various methods under ImageNet-C. As shown in Tab. 4, Tent achieves the fastest speed (110s) as it only performs entropy optimization on samples without additional operations. DDA requires significantly more time since it needs to use the source data to train additional networks. The sharpness-aware optimization in SAR and the data augmentation in DeYO require additional model forward or backward passes, resulting in more time cost. Our ReCAP achieves significantly superior performance with the second-best time cost (just +5% compared to Tent), highlighting its efficiency.

## 6. Ablation Study and Visualization

In this section, without loss of generality, we conduct ablation studies and visualizations on ResNet in label shift scenes for the sake of brevity. Focusing on two pivotal components of ReCAP, *Probabilistic region modeling* and *Efficient metric of region confidence*, we perform various experiments to analyze their impacts. Refer to the Appendix C for more experiments and analysis.

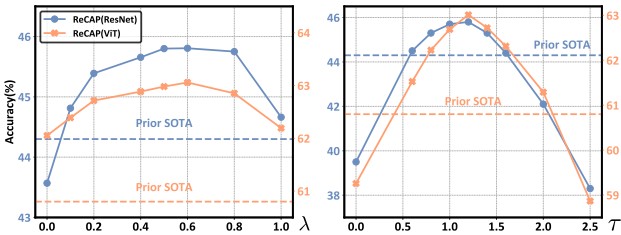

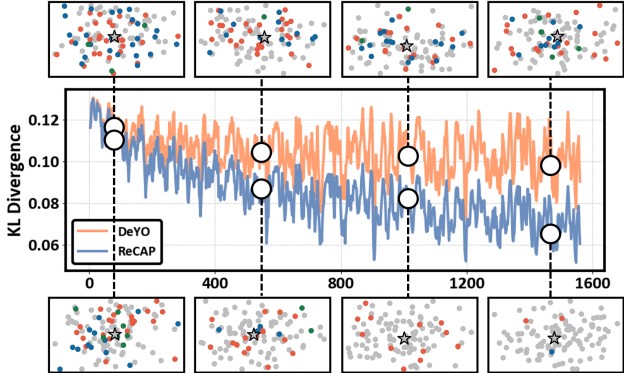

*Figure 4.* (a) Performance with varying strengths $\lambda$ of the variance term. (b) Performance with different ranges $\tau$ of the local region.

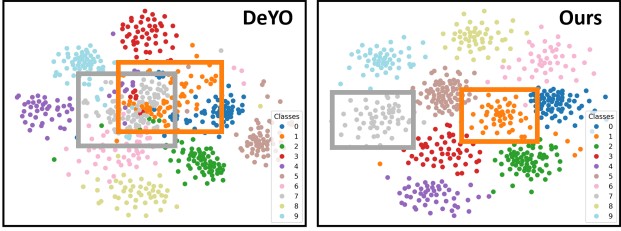

*Figure 5.* The t-SNE visualization of feature space in Snow corruption type after model adaptation.

### 6.1. Hyper-parameter Robustness

There are two important hyper-parameters in our method: the coefficient $\lambda$, which determines the trade-off of bias term and variance term, and the coefficient $\tau$, which controls the scope of the local region. We conduct ablation experiments on these two key coefficients independently:

As shown in Fig. 4(a), the different strengths of variance term yields stable performance gains. However, when $\lambda$ exceeds the optimal range, model updates may prioritize consistency over confidence optimization, leading to worse performance. To balance the effects of two terms in region confidence, we set $\lambda$ to 0.5 by default. In Fig. 4(b), performance improves as the region scale increases, peaking at $\tau = 1.2$. However, when $\tau$ exceeds a reasonable range (e.g., 2.5), the local region may suffer from category mixing, leading to performance degradation. Therefore, we set $\tau$ to 1.2 by default. Within the range of $\tau \in [0.5, 1.5]$, our method consistently outperforms prior SOTA methods, demonstrating the robustness of the region scale in our approach.

### 6.2. Class Separability after Adaptation

To analyze the effects of WTTA methods on feature representations, we visualize the feature representation of different categories after model adaptation using t-SNE (Van der Maaten & Hinton, 2008) in Fig. 5. Compared to the latest SOTA DeYO, our ReCAP exhibits more compact intra-class features and more distinct inter-class separability (e.g., class 1 in orange and class 7 in grey). Since our method facilitates a more efficient transfer process, the detrimental effects of distribution shifts on the clustering properties of the feature space can be quickly alleviated, yielding representations with clear classification boundaries in testing data.

*Figure 6.* Visualization of prediction results and KL Divergence of prediction probabilities in local region. Different colors of points represent different predicted classes of samples within the region.

### 6.3. Prediction Consistency Comparison

To analyze the effect on local consistency, we sample 128 points within the local distribution of each sample to assess fluctuations in prediction probabilities during the adaptation process. As shown in Fig. 6, compared to DeYO, our approach consistently exhibits significantly lower KL Divergence values throughout the entire optimization process. Furthermore, we visualize the prediction results during the training process. The visualization reveals that the prior method fails to ensure consistent predictions, whereas our method progressively enhances consistency, effectively reducing the dispersion in training dynamics.

## 7. Conclusion

In this paper, we propose ReCAP, a novel method designed to capture consistent optimization dynamics that are often disrupted by entropy minimization, offering an effective solution for leveraging the region-integrated training. We construct a novel training objective to replace vanilla entropy and further develop an asymptotic analysis framework to derive a more practical and flexible proxy for efficient training in TTA. We demonstrate its consistent effectiveness across a wide range of shift types, challenging wild scenarios, and diverse model architectures. We hope that our work will inspire future research move beyond the focus on individual data points, exploring more effective ways to leverage regional knowledge for robust and efficient adaptation.

**Limitations.** First, our evaluations focus on classification benchmarks. While the impact of Wild TTA in other tasks remains underexplored in existing research, it's crucial for understanding the capability boundary of methods. In future work, we plan to establish a more comprehensive benchmark across various tasks for a broader evaluation. Second, the local region we design is domain-wise, *i.e.* the same shape to all samples within a domain. However, given the differences in class cluster boundaries or the distance to boundaries, the region should ideally vary at the class-wise or sample-wise level. In future work, we will explore more fine-grained region modeling mechanisms to address this limitation.

## Acknowledgements

This work was supported by the Program of Beijing Municipal Science and Technology Commission Foundation (No.Z241100003524010), in part by the National Natural Science Foundation of China under Grant 62088102, in part by AI Joint Lab of Future Urban Infrastructure sponsored by Fuzhou Chengtou New Infrastructure Group and Boyun Vision Co. Ltd, and in part by the PKU-NTU Joint Research Institute (JRI) sponsored by a donation from the Ng Teng Fong Charitable Foundation.

## Impact Statement

This paper presents work whose goal is to advance the field of Test-Time Adaptation. Its societal impact lies primarily in its potential to expand the applicability of machine learning models in real-world complex scenarios. By improving the stability and efficiency of model adaptation, our work facilitates model deployment on edge devices where storage and computational power are limited. Ethically, it promotes more sustainable machine learning practices by reducing computational overhead, which in turn lowers energy consumption. This aligns with broader environmental sustainability goals and supports the development of more efficient, eco-friendly AI technologies for various industries.

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

# Beyond Entropy: Region Confidence Proxy for Wild Test-Time Adaptation

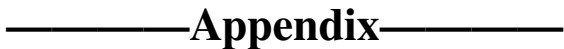

The structure of Appendix is as follows:

- Appendix A contains all missing proofs in the main manuscript.

- Appendix B presents additional experimental results on supplementary datasets.

- Appendix C provides further ablation studies and visualizations.

- Appendix D details the datasets and the methods used for comparison.

- Appendix E expands on related work in relevant fields.

## A. Theoretical Proof

Below, we will provide detailed proofs of the theoretical results presented in the methodology Sec. 4.2.

**Notation.** First, we recall the notation that we used in the main paper as well as this appendix: $C$ denotes the number of classes. $F_\theta$ denotes the model and $\theta$ denotes the model parameters. $x$ denotes a testing sample and $z \in R^d$ denotes its corresponding feature. $\mathbb{E}$ denotes the operation of expectation. $\mathcal{N}(\mu, \Sigma)$ denotes the Gaussian distribution with a mean of $\mu$ and a covariance of $\Sigma$. The subscript $(\cdot)_i$ denotes the $i$-th dimension, corresponding to the $i$-th class. $A \in \mathbb{R}^{C \times d}$ and $b \in \mathbb{R}^C$ denote the linear and bias coefficients of the affine layer in the classifier, with $a_i$ and $b_i$ being their $i$-th dimensions, respectively. $p_\theta(z)_i = (\text{softmax}\,(Az + b))_i = \frac{e^{a_i \cdot z + b_i}}{\sum_{j=1}^C e^{a_j \cdot z + b_j}}$ denotes the predicted probability of sample $x$ belonging to i-th class. $\mathcal{L}_{ent}(x) = -p_\theta(z) \cdot \log p_\theta(z) = -\sum_{i=1}^C p_\theta(z)_i \log p_\theta(z)_i$ denotes the prediction entropy of the sample $x$ and its corresponding feature $z$.

### A.1. Two Lemma Inequalities

Subsequently, we provide the proof of two important inequalities that we need to use to derive the conclusion of Proposition 4.3 & 4.4 in the main paper.

**Lemma A.1.** *(Bias Term under Finite Sampling) Given N features $z_1, \ldots, z_N$ drawn from the local region and their corresponding probabilities: $p_\theta(z_1), \ldots, p_\theta(z_N)$. The bias term on these features satisfies the following inequality:*

$$\sum_{i=1}^N \mathcal{L}_{ent}(p_\theta(z_i)) \leqslant -\sum_{i=1}^C \frac{\sum_{k=1}^N p_\theta(z_k)_i}{N} \cdot \left(\sum_{j=1}^N \log p_\theta\,(z_j)_i\right). \tag{10}$$

*Proof.* We begin by examining the difference between the left-hand side (LHS) and the right-hand side (RHS) of the inequality. By merging identical logarithmic terms, we can reformulate the expression into multiple summations, which we then simplify using the commutative property of summation:

$$\begin{aligned}
RHS - LHS &= \sum_{i=1}^N \sum_{j=1}^C \left(p_\theta(z_i)_j - \frac{1}{N}\sum_{k=1}^N p_\theta(z_k)_j\right) \log p_\theta(z_i)_j \\
&= \frac{1}{N} \sum_{i=1}^N \sum_{j=1}^C \sum_{k=1}^N (p_\theta(z_i)_j - p_\theta(z_k)_j) \log p_\theta(z_i)_j.
\end{aligned} \tag{11}$$

Since $C$ dimensions of the probability in Eq. 11 are independent of each other, we can treat each dimension separately. Therefore, it suffices to prove the following inequality for each dimension:

$$\frac{1}{N} \sum_{i=1}^N \sum_{k=1}^N \left(p_\theta\,(z_i)_j - p_\theta\,(z_k)_j\right) \log p_\theta\,(z_i)_j \geqslant 0. \tag{12}$$

Applying Fubini's theorem allows us to interchange the order of summation in Eq. 12. We also interchange the indices $i$ and $k$ to obtain the following identity:

$$\sum_{i=1}^{N}\sum_{k=1}^{N}\left(p_\theta\left(z_i\right)_j - p_\theta\left(z_k\right)_j\right)\log p_\theta\left(z_i\right)_j = \sum_{k=1}^{N}\sum_{i=1}^{N}\left(p_\theta\left(z_i\right)_j - p_\theta\left(z_k\right)_j\right)\log p_\theta\left(z_i\right)_j$$
$$= \sum_{i=1}^{N}\sum_{k=1}^{N}\left(p_\theta\left(z_k\right)_j - p_\theta\left(z_i\right)_j\right)\log p_\theta\left(z_k\right)_j . \tag{13}$$

We notice that the results in the first and third lines in Eq. 13 differ slightly, with the only variation being in the logarithm of the probability. Therefore, we replace the original expression with the average of these two terms and combine the common terms into the form of a product of two differences:

$$\frac{1}{N}\sum_{i=1}^{N}\sum_{k=1}^{N}\left(p_\theta\left(z_i\right)_j - p_\theta\left(z_k\right)_j\right)\log p_\theta\left(z_i\right)_j$$
$$= \frac{1}{2}\left(\sum_{i=1}^{N}\sum_{k=1}^{N}\left(p_\theta\left(z_i\right)_j - p_\theta\left(z_k\right)_j\right)\log p_\theta\left(z_i\right)_j + \sum_{i=1}^{N}\sum_{k=1}^{N}(p_\theta\left(z_k\right)_j - p_\theta\left(z_i\right)_j)\log p_\theta\left(z_k\right)_j\right) \tag{14}$$
$$= \frac{1}{2N}\sum_{i=1}^{N}\sum_{k=1}^{N}\left(p_\theta\left(z_i\right)_j - p_\theta\left(z_k\right)_j\right)(\log p_\theta\left(z_i\right)_j - \log p_\theta\left(z_k\right)_j).$$

Since $p_\theta\left(z_i\right)_j - p_\theta\left(z_k\right)_j$ and $\log p_\theta\left(z_i\right)_j - \log p_\theta\left(z_k\right)_j$ have the same sign, the product of these two terms is non-negative:

$$\left(p_\theta\left(z_i\right)_j - p_\theta\left(z_k\right)_j\right)\left(\log p_\theta\left(z_i\right)_j - \log p_\theta\left(z_k\right)_j\right) \geqslant 0. \tag{15}$$

Combining Eq. 14 and Eq. 15, we conclude that the inequality in Eq. 12 holds. Summing over all $C$ dimensions yields the desired result:

$$\sum_{i=1}^{N}\mathcal{L}_{ent}(p_\theta(z_i)) \leqslant -\sum_{i=1}^{C}\frac{\sum_{k=1}^{N}p_\theta(z_k)_i}{N}\cdot\left(\sum_{j=1}^{N}\log p_\theta\left(z_j\right)_i\right). \tag{16}$$

$$\square$$

**Lemma A.2.** *(Upper Bound of Negative Log-Likelihood) Given a feature $z$ and its local region $\Omega$ which follows a Gaussian distribution $\mathcal{N}(\mu,\Sigma)$. The expected value of the logarithm of the predicted probability for the $i$-th class satisfies the following inequality:*

$$-\mathbb{E}_{z\sim\mathcal{N}(\mu,\Sigma)}\log p_\theta(z)_i \leq \log\sum_{j=1}^{C}e^{(a_j-a_i)\cdot\mu+(b_j-b_i)+\frac{1}{2}(a_j-a_i)\Sigma(a_j-a_i)^\top}, \tag{17}$$

*where the superscript $(\cdot)^\top$ denotes the transpose operation.*

*Proof.* First, we transform the left-hand side (LHS) of the inequality to eliminate the fraction, which complicates scaling. We rewrite it as follows:

$$LHS = \mathbb{E}_{z\sim\mathcal{N}(\mu,\Sigma)}\log\sum_{j=1}^{C}e^{(a_j-a_i)\cdot z+(b_j-b_i)}. \tag{18}$$

Since the logarithm function is concave (*i.e.*, $\log(x)'' = -\frac{1}{x^2}<0$), we can apply Jensen's inequality and the additivity of expectations to derive the following result:

$$LHS \leqslant \log\mathbb{E}_{z\sim\mathcal{N}(\mu,\Sigma)}\sum_{j=1}^{C}e^{(a_j-a_i)\cdot z+(b_j-b_i)} = \log\sum_{j=1}^{C}\mathbb{E}_{z\sim\mathcal{N}(\mu,\Sigma)}e^{(a_j-a_i)\cdot z+(b_j-b_i)}. \tag{19}$$

Through leveraging the moment property of the Gaussian distribution $\mathbb{E}_{X \sim \mathcal{N}(\mu, \sigma^2)} e^X = e^{\mu + 1/2\sigma^2}$ and noting that $a_i \cdot z + b_i \sim \mathcal{N}\left(a_i \cdot \mu + b_i, a_i \Sigma a_i^\top\right)$, we can directly compute the expectation in Eq. 19:

$$\log \sum_{j=1}^{C} \mathbb{E}_{z \sim \mathcal{N}(\mu, \Sigma)} e^{(a_j - a_i) \cdot z + (b_j - b_i)} = \log \sum_{j=1}^{C} e^{(a_j - a_i) \cdot \mu + (b_j - b_i) + \frac{1}{2}(a_j - a_i)\Sigma(a_j - a_i)^\top}. \tag{20}$$

Combining Eq. 19 and Eq.20, we obtain the inequality that needs to be proved:

$$-\mathbb{E}_{z \sim \mathcal{N}(\mu, \Sigma)} \log \frac{e^{a_i \cdot z + b_i}}{\sum_{j=1}^{C} e^{a_j \cdot z + b_j}} \leq \log \sum_{j=1}^{C} e^{(a_j - a_i) \cdot \mu + (b_j - b_i) + \frac{1}{2}(a_j - a_i)\Sigma(a_j - a_i)^\top}. \tag{21}$$

$\square$

### A.2. Closed-form Upper Bound

Finally, we utilize the two inequalities derived above to obtain the crucial results in this paper, *Regional Entropy*, which provides a closed-form upper bound for the expectation of the entropy loss over the local distribution, and *Regional Instability*, which offers a closed-form upper bound for the expectation of the KL divergence between the prediction probability distribution over the distribution and the original prediction at its center.

**Proposition A.3.** *(Efficient Metric of Bias Term) Given a feature $z$ and its local region $\Omega$ which follows a Gaussian distribution $\mathcal{N}(\mu, \Sigma)$. The expectation of entropy loss over the entire distribution has a closed-form upper bound:*

$$\mathbb{E}_{\Omega}[\mathcal{L}_{ent}] = -\mathbb{E}_{\tilde{z} \sim \mathcal{N}(z, \Sigma)} \sum_{i=1}^{C} p_\theta(\tilde{z})_i \log p_\theta(\tilde{z})_i$$

$$\leq \sum_{j=1}^{C} \frac{e^{a_j \cdot z + b_j + \frac{1}{2} a_j \sum a_j^\top}}{\sum_{k=1}^{C} e^{a_k \cdot z + b_k + \frac{1}{2} a_k \sum a_k^\top}} \log \sum_{i=1}^{C} e^{(a_i - a_j) \cdot z + (b_i - b_j) + \frac{1}{2}(a_i - a_j)\Sigma(a_i - a_j)^\top} \triangleq \mathcal{L}_{RE}. \tag{22}$$

*where the upper bound $\mathcal{L}_{RE}$ is called **Regional Entropy**.*

*Proof.* First, by the definition of expectation, we can estimate the expectation of entropy using an infinite number of sampling $\tilde{z}_1, \tilde{z}_2, \ldots, \tilde{z}_N, \ldots \overset{i.i.d}{\sim} \mathcal{N}(z, \Sigma)$:

$$\mathbb{E}[\mathcal{L}_{ent}] = -\lim_{N \to +\infty} \frac{1}{N} \sum_{i=1}^{N} \sum_{j=1}^{C} p_\theta(\tilde{z}_i)_j \log p_\theta(\tilde{z}_i)_j. \tag{23}$$

Using Lemma A.1, we can bound the values of $p_\theta(\tilde{z}_i)_j, i = 1, 2, \ldots, N$ in Eq. 23, by their mean. This gives us the following inequality:

$$\mathbb{E}[\mathcal{L}_{ent}] \leq -\lim_{N \to +\infty} \frac{1}{N} \sum_{i=1}^{N} \sum_{j=1}^{C} \frac{\sum_{k=1}^{N} p_\theta(z_k)_j}{N} \log p_\theta(\tilde{z}_i)_j. \tag{24}$$

Next, we use the expectation to approximate $\frac{\sum_{k=1}^{N} p_\theta(z_k)_j}{N}$ as $N$ approaches infinity. To ensure the integrability, we first take the expectation and then apply the softmax operation. This can be derived from Eq. 20 as follows:

$$\overline{p_\theta(z)}_i := \frac{\mathbb{E}_{\tilde{z} \sim \mathcal{N}(z, \Sigma)} e^{a_i \cdot \tilde{z} + b_i}}{\mathbb{E}_{\tilde{z} \sim \mathcal{N}(z, \Sigma)} \sum_{j=1}^{C} e^{a_j \cdot \tilde{z} + b_j}} = \frac{e^{a_i \cdot z + b_i + \frac{1}{2} a_i \Sigma a_i^\top}}{\sum_{j=1}^{C} e^{a_j \cdot z + b_j + \frac{1}{2} a_j \Sigma a_j^\top}}. \tag{25}$$

Through the definition of the limit, we have that for any $\epsilon \geq 0$, there exists a positive integer $N_0$ such that for any $N \geq N_0$, the following inequality holds:

$$\left| \frac{\sum_{k=1}^{N} p_\theta(z_k)_j}{N} - \overline{p_\theta(z)}_j \right| \leq \epsilon. \tag{26}$$

Combining Eq. 24 and Eq. 26, and substituting the specific value of $\overline{p_\theta(z)}_j$ from Eq. 25, we have:

$$
\mathbb{E}[\mathcal{L}_{ent}] \leqslant - \lim_{N \to +\infty} \frac{1}{N} \sum_{i=1}^{N} \sum_{j=1}^{C} \left( \overline{p_\theta(z)}_j + \epsilon \right) \log p_\theta(\tilde{z}_i)_j
$$

$$
= - \lim_{N \to +\infty} \frac{1}{N} \sum_{i=1}^{N} \sum_{j=1}^{C} \left( \frac{e^{a_j \cdot z + b_j + \frac{1}{2} a_j \Sigma a_j^\top}}{\sum_{k=1}^{C} e^{a_k \cdot z + b_k + \frac{1}{2} a_k \Sigma a_k^\top}} + \epsilon \right) \log p_\theta(\tilde{z}_i)_j.
$$

(27)

Through the discrete form of Fubini's theorem, we can exchange the order of summation and extract terms that are independent of the limit calculation:

$$
- \lim_{N \to +\infty} \frac{1}{N} \sum_{i=1}^{N} \sum_{j=1}^{C} \left( \frac{e^{a_j \cdot z + b_j + \frac{1}{2} a_j \Sigma a_j^\top}}{\sum_{k=1}^{C} e^{a_k \cdot z + b_k + \frac{1}{2} a_k \Sigma a_k^\top}} + \epsilon \right) \log p_\theta(\tilde{z}_i)_j
$$

$$
= \sum_{j=1}^{C} \left( \frac{e^{a_j \cdot z + b_j + \frac{1}{2} a_j \Sigma a_j^\top}}{\sum_{k=1}^{C} e^{a_k \cdot z + b_k + \frac{1}{2} a_k \Sigma a_k^\top}} + \epsilon \right) \lim_{N \to +\infty} - \frac{1}{N} \sum_{i=1}^{N} \log p_\theta(\tilde{z}_i)_j.
$$

(28)

Through the definition of the expectation and Lemma. A.2, we have:

$$
- \frac{1}{N} \sum_{i=1}^{N} \log p_\theta(\tilde{z}_i)_j = - \mathbb{E}_{\tilde{z} \sim \mathcal{N}(z,\Sigma)} \log p_\theta(\tilde{z})_j
$$

$$
\leq \log \sum_{i=1}^{C} e^{(a_i - a_j) \cdot z (b_i - b_j) + \frac{1}{2}(a_i - a_j)\Sigma(a_i - a_j)^\top}.
$$

(29)

Combining Eq. 27, Eq. 28 and Eq. 29, we have:

$$
\mathbb{E}[\mathcal{L}_{ent}] \leqslant \sum_{j=1}^{C} \left( \frac{e^{a_j \cdot z + b_j + \frac{1}{2} a_j \sum a_j^\top}}{\sum_{k=1}^{C} e^{a_k \cdot z + b_k + \frac{1}{2} a_k \sum a_k^\top}} + \epsilon \right) \log \sum_{i=1}^{C} e^{(a_i - a_j) \cdot z}
$$

$$
\cdot e^{(b_i - b_j) + \frac{1}{2}(a_i - a_j)\Sigma(a_i - a_j)^\top}, \quad \text{for } \forall \epsilon \geq 0.
$$

(30)

Taking $\epsilon \to 0$ in Eq. 30, we obtain the inequality we need to prove in Eq. 22. $\qquad \square$

**Proposition A.4.** *(Efficient Metric of Variance Term) Given a feature $z$ and its local region $\Omega$ which follows a Gaussian distribution $\mathcal{N}(\mu, \Sigma)$. The expectation of Kullback-Leibler divergence between the output probability over this distribution and the probability at its center has a upper bound:*

$$
E_{\tilde{z} \sim \mathcal{N}(z,\Sigma)} KL\left(p_\theta(z) \| p_\theta(\tilde{z})\right) \leq \sum_{j=1}^{C} \frac{e^{a_j \cdot z + b_j}}{\sum_{k=1}^{C} e^{a_k \cdot z + b_k}} \cdot \log \sum_{i=1}^{C} \frac{e^{a_i \cdot z + b_i}}{\sum_{k=1}^{C} e^{a_k \cdot z + b_k}} \cdot e^{\frac{1}{2}(a_i - a_j)\sum(a_i - a_j)^\top} \triangleq \mathcal{L}_{RI}, \quad (31)
$$

*where the upper bound $\mathcal{L}_{RI}$ is called **Regional Instability**.*

*Proof.* We first transform the left-hand side of the inequality into the following form:

$$
LHS = \mathbb{E}_{\tilde{z} \sim N(z,\Sigma)} \sum_{i=1}^{C} p_\theta(z)_i \log \frac{p_\theta(z)_i}{p_\theta(\tilde{z})_i}
$$

$$
= - \mathbb{E}_{\tilde{z} \sim N(z,\Sigma)} \sum_{i=1}^{C} p_\theta(z)_i \log \left( \frac{1}{p_\theta(z)_i} \cdot \frac{e^{a_i \cdot \tilde{z} + b_i}}{\sum_{j=1}^{C} e^{a_j \cdot \tilde{z} + b_j}} \right).
$$

(32)

Since $p_\theta(z)_i$ is independent of the expectation operation, by applying Lemma A.2, we have:

$$
LHS \leq \sum_{i=1}^{C} p_\theta(z)_i \cdot \log \left( p_\theta(z)_i \cdot \sum_{j=1}^{C} e^{(a_j - a_i) \cdot z + (b_j - b_i) + \frac{1}{2}(a_j - a_i)\Sigma(a_j - a_i)^\top} \right).
$$

(33)

Substituting the definition of $p_\theta(z)$, we have:

$$
\begin{aligned}
p_\theta(z)_i \cdot \sum_{j=1}^{C} & e^{(a_j-a_i)\cdot z+(b_j-b_i)+\frac{1}{2}(a_j-a_i)\Sigma(a_j-a_i)^\top} \\
&= \frac{e^{a_i\cdot z+b_i}}{\sum_{k=1}^{C} e^{a_k\cdot z+b_k}} \cdot \sum_{j=1}^{C} e^{(a_j-a_i)\cdot z+(b_j-b_i)+\frac{1}{2}(a_j-a_i)\Sigma(a_j-a_i)^\top} \\
&= \sum_{j=1}^{C} \frac{e^{a_j\cdot z+b_j}}{\sum_{k=1}^{C} e^{a_k\cdot z+b_k}} \cdot e^{\frac{1}{2}(a_j-a_i)\Sigma(a_j-a_i)^\top}.
\end{aligned}
\tag{34}
$$

Combining Eq. 23 and Eq. 24, we obtain the inequality we need to prove in Eq. 31. □

# B. Further Experiments

In this section, we broaden the scope of our investigation to evaluate the performance of our method across a variety of complex and diverse scenarios. To this end, we conduct experiments on Wild TTA scenarios using two challenging datasets: ImageNet-R (Hendrycks et al., 2021) and VisDA-2021 (Bashkirova et al., 2022). Both datasets present an array of distribution shifts and variations in data styles that extend beyond the typical corruptions found in ImageNet-C (Hendrycks & Dietterich, 2019), thereby providing a more comprehensive evaluation framework. By applying our method to these datasets, we examine its robustness under mixed testing domain scenarios, incorporating the cases with label shifts or batch size restricted to 1.

## B.1. Wild Scenes on ImageNet-R and VisDA-2021

We conduct additional experiments on WTTA scenarios using the ImageNet-R (Hendrycks et al., 2021) and VisDA-2021 (Bashkirova et al., 2022) datasets with ResNet and ViT architectures. These datasets are characterized by diverse distribution

| Model+Method | Limited Batch Size = 1 | | | Imbalanced Label Shift | | | Average |
|---|---|---|---|---|---|---|---|
| | ResNet | VitBase | Avg | ResNet | VitBase | Avg | |
| No-Adapt Model | 40.8 | 43.1 | 42.0 | 40.8 | 43.1 | 42.0 | 42.0 |
| • Tent (Wang et al., 2020) | 43.2 | 43.8 | 43.5 | 42.4 | 46.8 | 44.6 | 44.1 |
| • EATA (Niu et al., 2022) | 44.1 | 52.5 | 48.3 | 42.1 | 50.5 | 46.3 | 47.3 |
| • SAR (Niu et al., 2023) | 46.7 | 55.5 | 51.1 | 44.3 | 54.4 | 49.4 | 50.2 |
| • DeYO (Lee et al., 2024) | 48.1 | 59.2 | 53.7 | 46.7 | 58.5 | 52.6 | 53.1 |
| • ReCAP (Ours) | $\mathbf{51.5}_{\pm0.5}$ | $\mathbf{61.1}_{\pm0.6}$ | $\mathbf{56.3}_{\pm0.5}$ | $\mathbf{49.6}_{\pm0.2}$ | $\mathbf{60.4}_{\pm0.2}$ | $\mathbf{55.0}_{\pm0.2}$ | $\mathbf{55.7}_{\pm0.4}$ |

*Table 5.* Comparisons with state-of-the-art methods on **ImageNet-R** under **Limited Batch Size = 1** & **Imbalanced Label Shift** regarding Accuracy (%). We report mean&std over 3 independent runs. The best results are in bold.

| Model+Method | Limited Batch Size = 1 | | | Imbalanced Label Shift | | | Average |
|---|---|---|---|---|---|---|---|
| | ResNet | VitBase | Avg | ResNet | VitBase | Avg | |
| No-Adapt Model | 43.5 | 44.3 | 43.9 | 43.5 | 44.3 | 43.9 | 43.9 |
| • Tent (Wang et al., 2020) | 43.9 | 50.6 | 47.3 | 43.7 | 50.1 | 46.9 | 47.1 |
| • EATA (Niu et al., 2022) | 44.2 | 49.5 | 46.9 | 43.5 | 51.6 | 47.6 | 47.2 |
| • SAR (Niu et al., 2023) | 45.2 | 52.8 | 49.0 | 44.7 | 53.9 | 49.3 | 49.2 |
| • DeYO (Lee et al., 2024) | 45.8 | 58.5 | 52.2 | 45.2 | 57.1 | 51.2 | 51.7 |
| • ReCAP (Ours) | $\mathbf{48.0}_{\pm0.2}$ | $\mathbf{59.2}_{\pm0.9}$ | $\mathbf{53.6}_{\pm0.6}$ | $\mathbf{47.7}_{\pm0.2}$ | $\mathbf{58.5}_{\pm0.6}$ | $\mathbf{53.1}_{\pm0.4}$ | $\mathbf{53.4}_{\pm0.5}$ |

*Table 6.* Comparisons with state-of-the-art methods on **VisDA-2021** under **Limited Batch Size = 1** & **Imbalanced Label Shift** regarding Accuracy (%). We report mean&std over 3 independent runs. The best results are in bold.

shifts, including variations in data styles that extend beyond mere corruption. Consequently, for these two datasets, we consistently consider the mixed testing domain scenarios and incorporate cases with label shifts or batch size = 1. This rigorous testing environment ensures a comprehensive assessment of model robustness under real-world conditions. All evaluations are performed using the same implementation details as outlined in the main paper.

Tab. 5 presents the results on ImageNet-R for batch size = 1 and imbalanced label distribution shift scenarios. Consistent with the findings in the main paper for ImageNet-C, ReCAP demonstrates superior performance across various scenarios and architectures on the ImageNet-R benchmark. We also compare our ReCAP method with previous state-of-the-art approaches on the VisDA-2021 dataset. The results in Tab. 6 align with those observed on ImageNet-C and ImageNet-R, where ReCAP similarly exhibits the best performance across all Wild settings on VisDA-2021.

We further investigate the performance of our method under different distribution shifts. As discussed in the main paper, our ReCAP approach provides an efficient proxy to optimize region confidence, effectively reducing inconsistent predictions and enhancing global optimization efficiency. Compared to other sample selection WTTA methods, ReCAP consistently improves performance across various architectures and scenarios, achieving average performance gains of $+2.6\%$ and $+1.7\%$ on ImageNet-R and VisDA-2021, respectively. The experimental results further validate the generalizability of our method across different types of shifts, providing a more comprehensive understanding of its effectiveness.

## C. Additional Ablation Study and Visualization

In Section 6 of the main paper, we provided a comprehensive validation of the hyperparameter robustness of $\lambda$ and $\tau$, along with visualizations that illustrate the effects of ReCAP on class separability and local consistency during the adaptation process. In this section, we extend our analysis by further investigating the sensitivity of key parameters and the evolution of the model adaptation, providing additional insights into the effectiveness and robustness of our method.

### C.1. Sensitivity of $\tau_{RE}$ in ReCAP

The hyperparameter $\tau_{RE}$ plays a crucial role in determining the sample selection criterion within the ReCAP framework. To understand its impact on performance, we evaluate ReCAP under varying $\tau_{RE}$ values. As shown in Fig. 7, increasing $\tau_{RE}$ leads to the inclusion of more samples in the training process, which results in improved performance. The performance peaks at $0.8/1.0 \times \ln(C)$ for ResNet/ViT, respectively, indicating an optimal balance between sample inclusion and computational efficiency. However, when $\tau_{RE}$ exceeds this optimal range, the sample selection mechanism becomes too permissive, allowing for the inclusion of noisy or detrimental samples, which ultimately degrades performance. Despite this,

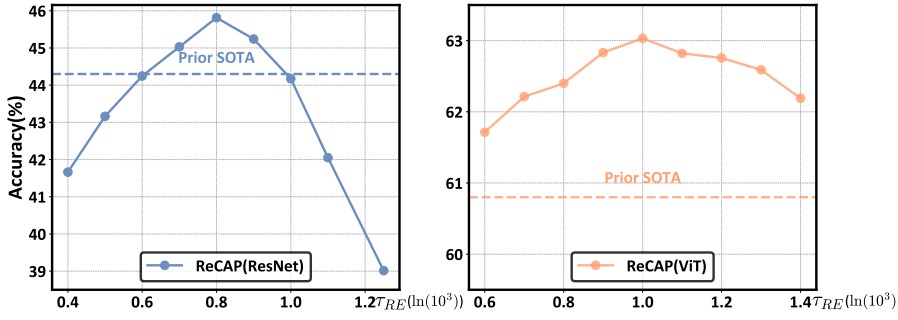

*Figure 7.* Performance under different selection boundary $\tau_{RE}$ for ResNet and ViT on ImageNet-C under label shifts.

*Table 7.* Effects of components in ReCAP. For a fair comparison, '+Vanilla Entropy' uses entropy-based selection and weighting.

| Component | | | Corruption Category | | | | Average |
|---|---|---|---|---|---|---|---|
| Vanilla Entropy | $\mathcal{L}_{RE}$ in Eq. 7 | $\mathcal{L}_{RI}$ in Eq. 8 | Noise | Blur | Weather | Digital | |
| ✔ | | | 24.9 | 19.6 | 41.5 | 37.8 | 31.4 |
| | ✔ | | 41.9 | 29.9 | 56.4 | 44.5 | 43.3 |
| ✔ | | ✔ | 41.3 | 26.8 | 55.0 | 47.3 | 42.7 |
| | ✔ | ✔ | **42.9** | **31.0** | **57.1** | **51.2** | **45.8** |

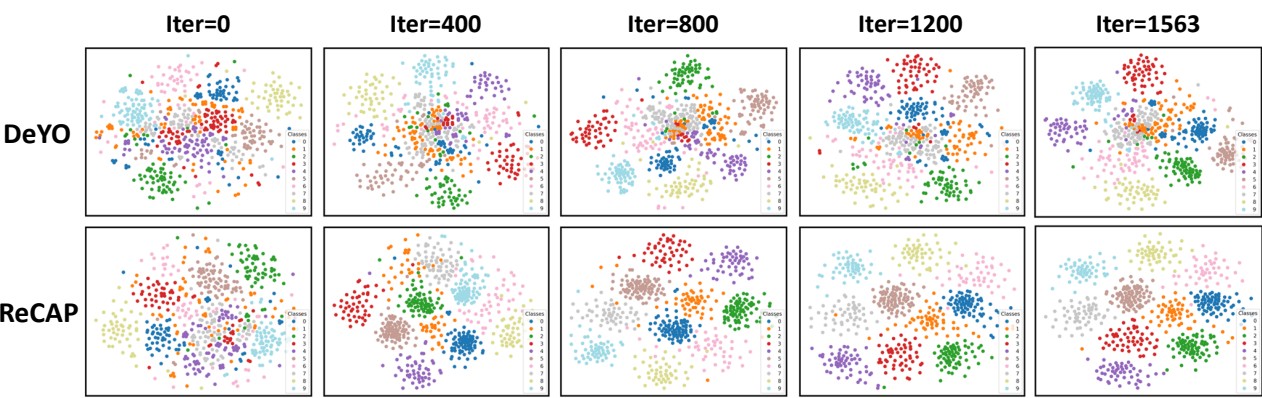

*Figure 8.* The evolution of feature space under DeYO and ReCAP methods. The visualizations are conducted on ImageNet-C under labl shift scenario with ResNet50.

ReCAP maintains a consistent performance advantage over prior state-of-the-art methods across a wide range of $\tau_{RE}$ values, showcasing its robustness to variations in the sample selection boundary.

### C.2. Effectiveness of Components in ReCAP

We investigate the impact of individual components within the ReCAP framework by comparing the full ReCAP method with variations that omit key parts of the approach. Specifically, we compare ReCAP to a vanilla entropy minimization strategy and systematically add back components to assess their contribution to performance. The results, as shown in Tab. 7, reveal that the region confidence achieves its best effect only when both components are included, with performance gains of +2.5% and +3.1%, respectively. This demonstrates the complementary nature of these components in enhancing adaptation performance. Overall, the full ReCAP method consistently delivers the best performance, further validating the compatible effects of its key components in improving adaptation across various scenarios.

### C.3. Evolution Process of Model Adaptation

To further validate the effectiveness of ReCAP in improving adaptation efficiency, we visualize the evolution of the model's feature space using t-SNE (Van der Maaten & Hinton, 2008). Fig. 8 illustrates the adaptation process for both ReCAP and the latest SOTA method, DeYO. Notably, ReCAP demonstrates superior adaptation efficiency by achieving better class separability throughout the adaptation process. At Iteration 800, ReCAP exhibits distinct, well-separated class clusters, even outperforming DeYO's final state (at Iteration 1563) in terms of class boundaries. This early emergence of clear class separability highlights the efficiency of our method in accelerating the adaptation process, ensuring that ReCAP achieves a more structured and organized feature space compared to DeYO. These visualizations not only reinforce the advantages of ReCAP in enhancing adaptation efficiency but also provide strong evidence of its effectiveness in real-world scenarios where quick and robust adaptation is critical.

## D. More Implementation Details

### D.1. Baseline Methods

We compare ReCAP with the following SOTA methods: MEMO (Zhang et al., 2022) enhances prediction consistency by leveraging multiple augmented copies of input samples, ensuring stable model outputs despite test data variations. Tent (Wang et al., 2020) reduces the entropy of test samples to guide model updates, driving the model to make more confident predictions.EATA (Niu et al., 2022) combines sample selection based on entropy with weighted adjustments to minimize entropy specifically for the selected samples. SAR (Niu et al., 2023) introduces sharpness awareness with entropy-based selection into the entropy minimization process, ensuring more stable adaptation in challenging wild scenarios. DeYO (Lee et al., 2024) prioritizes samples with dominant shape information and applies a dual selection criterion to identify more reliable samples for adaptation.

## D.2. More Details on Dataset

In this paper, we primarily evaluate the out-of-distribution (OOD) generalization ability of all methods using a widely adopted benchmark: ImageNet-C (Hendrycks & Dietterich, 2019). **ImageNet-C** is derived by applying a series of corruptions to the original ImageNet (Deng et al., 2009) test set, making it a large-scale benchmark for assessing model robustness under real-world distribution shifts. The dataset consists of 15 distinct types of corruptions, including Gaussian noise, shot noise, impulse noise, defocus blur, glass blur, motion blur, zoom blur, snow, frost, fog, brightness, contrast, elastic transformation, pixelation, and JPEG compression. Each corruption type is further categorized into five severity levels, with higher severity indicating more extreme perturbations and greater distribution shifts. These corruptions simulate real-world degradations that can occur in diverse environmental conditions, making ImageNet-C an essential tool for evaluating the resilience of models in challenging, real-world scenarios. As illustrated in Fig. 9, these corruptions span a broad spectrum, challenging the model to adapt to varied distortions of input images.

Additionally, we conduct experiments on two other challenging benchmarks, ImageNet-R (Hendrycks et al., 2021) and VisDA-2021 (Bashkirova et al., 2022), to further validate the robustness and adaptability of our method across different types of distribution shifts. **ImageNet-R** consists of 30,000 images representing artistic renditions of 200 classes from ImageNet, with each image showcasing various creative transformations, such as paintings, drawings, and sculptures, sourced from platforms like Flickr and curated through Amazon MTurk annotators. These artistic variations introduce unique challenges in terms of visual style, texture, and color distribution, which are notably different from the original ImageNet images. As shown in Fig. 10, these renditions demand the model to generalize beyond typical object recognition tasks and adapt to complex, non-photorealistic representations.

**VisDA-2021**, on the other hand, is a more diverse dataset that encompasses a broader range of domain shifts. It includes images from multiple sources such as ImageNet-O/R/C and ObjectNet (Barbu et al., 2019). The domain shifts in VisDA-2021 involve a variety of challenges, such as changes in artistic visual styles, textures, viewpoints, and corruptions. This diversity in shifts ensures a comprehensive evaluation of model performance under real-world conditions with large variations in object appearance and environmental factors.

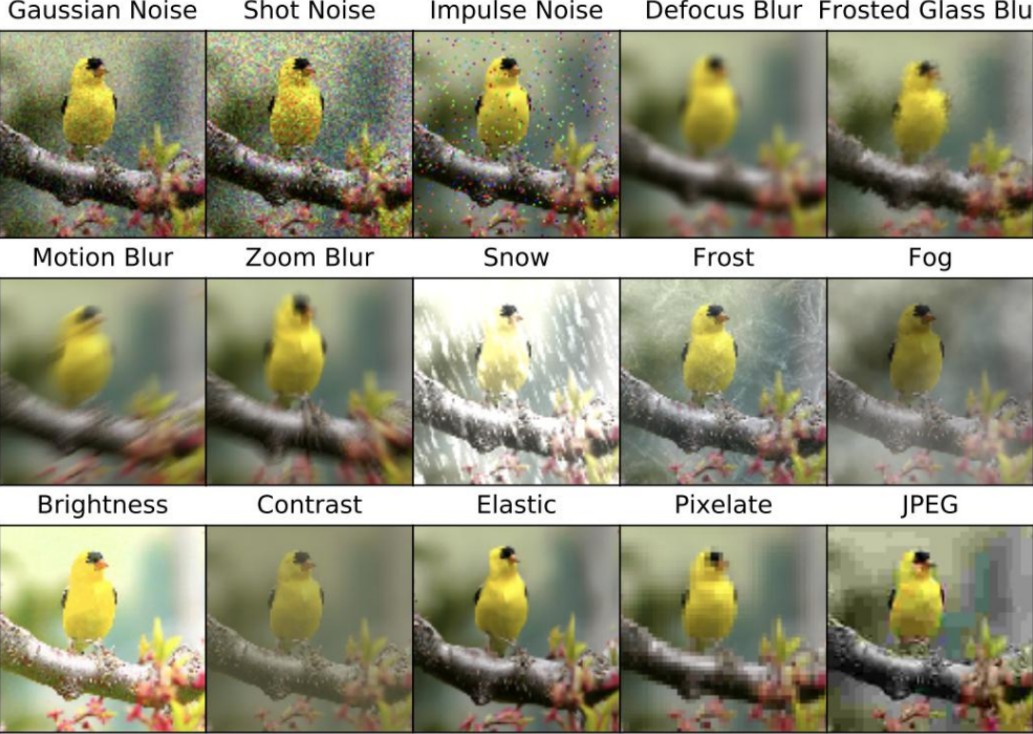

*Figure 9.* Visualizations of different corruption types in ImageNet corruption benchmark, which are taken from the original paper of ImageNet-C (Hendrycks & Dietterich, 2019).

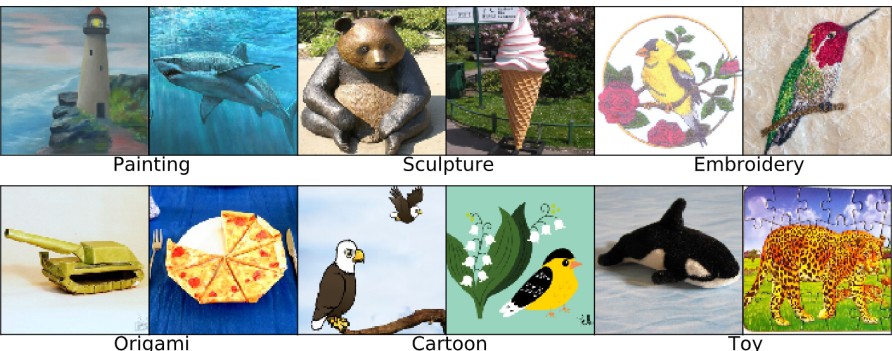

*Figure 10.* Visualizations of different style shift types in ImageNet-R benchmark, which are taken from the original paper of ImageNet-R (Hendrycks et al., 2021).

## E. Related Work

### E.1. Consistency Learning

Consistency learning is a key paradigm in semi-supervised learning (Berthelot et al., 2019), domain adaptation (Li et al., 2020; Araslanov & Roth, 2021), which enforces the model to produce stable and consistent predictions under different perturbations of the input data. It can be broadly categorized into two main approaches. First, consistency can be used as an effective criterion for identifying reliable samples (Prabhu et al., 2021; Yu et al., 2024b). This approach is based on the understanding that consistency under image transformations serves as a dependable indicator of model errors (Wei et al., 2020). For instance, methods like DeYO (Lee et al., 2024) select samples by evaluating the variation in pseudo-label probabilities under different augmentations, using this as a selection indicator.

Second, consistency learning can act as a regularization technique by introducing data augmentation (Sajjadi et al., 2016). By requiring the model to maintain consistent predictions across different augmentation variants of the same data, this approach enhances the model's robustness (Zhang et al., 2022; Xie et al., 2020). This technique has been widely used in semi-supervised learning and unsupervised domain adaptation. Unlike traditional regularization methods that rely on introducing augmented variants of the original samples, the proposed efficient proxy of *Region Confidence* in this work enhances local consistency directly from the features of the original samples. This eliminates the need for a lengthy process to obtain augmented variants, significantly improving the efficiency of optimizing consistency.

