# OpenReview forum: "Beyond Entropy: Region Confidence Proxy for Wild Test-Time Adaptation"
_ICML.cc/2025/Conference — ICML 2025 poster_

### Official Review · Reviewer_ynrW · 2025-03-04

**Overall Recommendation:** 4

**Summary:**

This paper introduces ReCAP, a novel TTA method based on local inconsistency of predictions.
Based on the finding that the local inconsistency increases and adaptation becomes difficult under wild distribution shifts, the region confidence is proposed as an alternative to entropy, a common objective in TTA.
Its finite-sample approximation is also derived to overcome the computational intractability of the original region confidence.
Experimental results show that ReCAP had higher accuracy on corrupted test data under wild settings (online, mixed shifts, and imbalanced labels).

## update after rebuttal
I appreciate the author's rebuttal and additional experiments. My concerns have been addressed. I have updated my score to 4.

**Claims And Evidence:**

Yes.

**Essential References Not Discussed:**

No.

**Experimental Designs Or Analyses:**

- Experimenting on continual TTA settings performed in recent TTA studies (e.g., EATA) would strengthen the efficacy of ReCAP in wild TTA settings.
- How was the sampling number from the region $N$ set? Examining the sensitivity of $N$ would be helpful.
- Comparing ReCAP with a simple Monte Carlo approximation of the original region confidence in Eq. (2) would make the proposed method more convincing.
- Ablation on the sample weighting and selection in Eq. (9) would be helpful.

**Methods And Evaluation Criteria:**

The motivation for using the proposed approximation of the region confidence in Eqs. (6) and (7) is unclear. One can use a simple Monte Carlo approximation. Providing some evidence that the proposed approximation is more efficient than Monte Carlo would be convincing.

**Other Comments Or Suggestions:**

N/A

**Other Strengths And Weaknesses:**

N/A

**Questions For Authors:**

N/A

**Relation To Broader Scientific Literature:**

The region confidence expands the commonly used sample-wise entropy.
It can improve existing entropy-based TTA methods.

**Theoretical Claims:**

I have checked the derivation of the region confidence.

---

> ### Author Rebuttal · Authors · 2025-03-31
>
> Thank you for taking the time to review our paper and providing valuable feedback. We would like to answer your questions below.
> >Q1: Experimenting on continual TTA settings performed in recent TTA studies (e.g., EATA) would strengthen the efficacy of ReCAP in wild TTA settings.
>
> A1: Thank you for your constructive suggestion to evaluate CTTA settings. We agree that such experiments would further strengthen the efficacy of our ReCAP. We conduct extensive experiments on CTTA for both classification and segmentation tasks.
>
> For classification, CTTA setup (Tab. 3 in response to Reviewer 6Q1i) and additional PTTA (CTTA + label shift)  setup (Tab. 3 in response to Reviewer eZDc) demonstrate that ReCAP consistently outperforms prior methods. For segmentation, results in Tab. 1 (this response) further confirm that ReCAP maintains robust adaptation performance in continual scenarios. These results underscore the broad applicability of ReCAP across continual and wild TTA settings.
>
> Table 1: Semantic segmentation results (mIoU) on the Cityscapes-to-ACDC CTTA setup based on the Segformer-B5 architecture.
> |Condition|Fog|Night|Rain|Snow|Fog|Night|Rain|Snow|Fog|Night|Rain|Snow|Fog|Night|Rain|Snow|Avg|
> |-|:---:|:---:|:---:|:---:|:---:|:---:|:---:|:---:|:---:|:---:|:---:|:---:|:---:|:---:|:---:|:---:|:---:|
> |Source|69.1|40.3|59.7|57.8|69.1|40.3|59.7|57.8|69.1|40.3|59.7|57.8|69.1|40.3|59.7|57.8|56.7|
> |TENT|69|40.2|60.1|57.3|66.5|36.3|58.7|54|64.2|32.8|55.3|50.9|61.8|29.8|51.9|47.8|52.3|
> |EATA|69.1|40.5|59.8|58.1|69.3|41.8|60.1|58.6|68.8|42.5|59.4|57.9|67.9|42.8|57.7|56.3|57.0|
> |CoTTA|70.9|41.1|62.4|59.7|70.8|40.6|62.7|59.7|70.8|40.5|62.6|59.7|70.8|40.5|62.7|59.7|58.4|
> |SAR|62.2|37.7|55.5|53.0|64.6|39.3|56.8|53.9|65.7|39.0|58.1|55.0|66.1|38.0|59.1|55.3|53.7|
> |Ours|72.7|43.8|63.9|61.1|71.9|42.2|64.1|60.1|71.0|40.5|63.5|58.8|70.3|39.3|62.8|57.2|59.0|
> >Q2: Sampling Number $N$ Sensitivity.
>
> A2: We appreciate your question and apologize for any confusion. Our method does not need any sampling due to the finite-to-infinite approximation in Propositions 4.3 and 4.4. Therefore, our method is entirely unaffected by the value of $N$. We will provide additional clarifications in the revised version to enhance clarity.
> >Q3: Comparing ReCAP with a simple Monte Carlo approximation of the original region confidence in Eq. (2) would make the proposed method more convincing.
>
> A3: We sincerely appreciate your valuable suggestion. We conduct a comprehensive comparison with the Monte Carlo (MC) approximation using different sampling numbers. As shown in Tab. 2, while MC provides a direct estimate of region confidence,  its accuracy is highly sensitive to the number of samples, leading to increased variance and a computational cost that scales linearly with the sample size.
>
> In contrast, ReCAP achieves significantly higher accuracy with lower variance during adaptation, demonstrating its superior stability and efficiency. These results further reinforce the motivation behind our finite-to-infinite approximation. We will incorporate this comparison into the revised version.
>
> Table 2: Comparison between the MC approximation and our finite-to-infinite approximation under 3 independent runs.
> | |ReCAP|MC (4)|MC (16)|MC (64)|MC (128)|
> |-|:---:|:---:|:---:|:---:|:---:|
> |Sampling number|NA|4|16|64|128|
> |Average Accuracy|42.2|31.7|34.8|38.7|40.7|
> |Standard deviation|0.1|9.8|3.7|1.4|0.3|
> |Running Time(s)|116|125|163|278|454|
> >Q4: Ablation on the sample weighting and selection in Eq. (9) would be helpful.
>
> A4: We appreciate the reviewer's suggestion and have conducted an additional ablation study to analyze the impact of different sample selection and weighting strategies. As shown in Tab. 3, the results lead to the following key observations:
> 1. Our region-based confidence optimization consistently enhances performance, surpassing the previous SOTA even when combined with the simplest entropy-based selection and weighting strategies.
> 2. The combination of our proposed selection and weighting achieves the best overall accuracy, further validating the effectiveness of our design.
>
> Table 3: Ablation study on selection and weighting strategies.
> |w/o selection|Entropy selection|Our selection|w/o weighting|Entropy weighting|Our weighting|ReCAP Accuracy|
> |:-:|:-:|:-:|:-:|:-:|:-:|:-:|
> |✓|||✓|||36.0|
> ||✓||✓|||38.9|
> |||✓|✓|||43.2|
> ||✓|||✓||44.9|
> |||✓||✓||45.7|
> |||✓|||✓|47.2|

---

> > ### Comment · Reviewer_ynrW · 2025-04-02
> >
> > I appreciate the author's rebuttal and additional experiments. My concerns have been addressed. I will update my score to 4.

---

> > > ### Author Response · Authors · 2025-04-02
> > >
> > > We are glad to know that our response has addressed your questions.
> > >
> > > We sincerely thank you for your thoughtful and constructive feedback. Through further discussions and experiments, we were able to more clearly communicate the contributions of our work.
> > >
> > > Again, we would like to thank you for appreciating our work and recognizing our contributions!
> > >
> > > Best,
> > >
> > > The Authors

---

### Official Review · Reviewer_eZDc · 2025-03-10

**Overall Recommendation:** 4

**Summary:**

This paper introduces a new Test-Time Adaptation Method (TTA) to combat domain shifts appearing at test time in extreme scenarios. In particular, it proposes ReCAP, a method that optimizes two terms: a bias term resembling a regional entropy around a given test data, and a variance term to enhance the consistency of the prediction of the model under neighboring features. Experiments are carried out on standard TTA benchmarks yielding consistent performance gain.

**Claims And Evidence:**

Yes

**Essential References Not Discussed:**

I think the paper did a good job relating itself to other related works.

**Experimental Designs Or Analyses:**

I checked the experimental sections in the paper and they all seem relevant, consistent with earlier work, and providing supportive results.

**Methods And Evaluation Criteria:**

Yes

**Other Comments Or Suggestions:**

Please refer to the "Other Strengths and Weaknesses" Section.

**Other Strengths And Weaknesses:**

While I am generally very positive about this paper, the following experiments I think are missing and would strengthen the paper.

1) Ablating $\mathcal L_0$: I checked the ablation experiments and did not find the one ablating the impact of $\mathcal L_0$. Further, when the proposed method is combined with SAR, is the data point selection mechanism of SAR employed or the proposed one?

2) In the efficiency comparison in Table 4: The proposed ReCAP computes more backward passes than EATA, however it is still more efficient in runtime. This seems a bit contradictory and deserves more discussion along with comparison against the more efficient variant of EATA (i.e. ETA). It is also important, given the efficiency of ReCAP, to show the performance gain under computational budgeted evaluation [A].

3) One extra [Optional] experiment is to extend the evaluation to the Practical TTA setting [B] which is closely related to the wild TTA setting.

[A] Evaluation of test-time adaptation under computational time constraints, ICML 2024

[B] Robust test-time adaptation in dynamic scenarios, CVPR 2023

**Questions For Authors:**

Please refer to the previous sections

**Relation To Broader Scientific Literature:**

This paper does a good job in linking their main contributions to earlier works. They further show experimentally how they can combine their proposed method with previous state-of-art showing further performance gain.

**Theoretical Claims:**

I skimmed through the proofs in the Appendices and they seem correct

---

> ### Author Rebuttal · Authors · 2025-03-31
>
> We deeply appreciate your positive comments and constructive suggestions on improving our paper. We will address your questions below.
> >Q1: I checked the ablation experiments and did not find the one ablating the impact of $\mathcal{L}_0$.
>
> A1: Due to space constraints, we provide the ablation study on $\mathcal{L}_0$ in Appendix C.1. As shown in Appendix Fig. 7, ReCAP consistently enhances performance across a broad range of $\mathcal{L}_0$ values, demonstrating its robustness to different selection boundaries. This result confirms that ReCAP does not rely on precise tuning of $\mathcal{L}_0$ and remains effective across varying settings. To improve accessibility, we will incorporate this ablation study into the main paper.
> >Q2: When ReCAP is combined with SAR, is the data selection mechanism of SAR employed or the proposed one?
>
> A2: When integrating ReCAP with SAR, we replace the original entropy selection with our proposed strategy, allowing for a direct evaluation of ReCAP's effectiveness. Likewise, when combining ReCAP with DeYO, we follow the same replacement strategy. We will clarify this in the revised version to eliminate any ambiguity.
> >Q3: Efficiency comparison with EATA and ETA.
>
> A3: Thank you for raising this point. While EATA performs fewer backward passes on test samples, it requires additional computation for Fisher regularization on extra source samples, resulting in a higher runtime compared to ReCAP.
>
> For comparison with ETA, we provide additional evaluations in Tab. 1. Although ETA offers a slight runtime improvement, it struggles to adapt to dynamic shifts in wild TTA scenarios. In contrast, ReCAP effectively balances efficiency and performance, achieving superior accuracy with marginal additional computation cost.
>
> Table 1: Running time for 50,000 images and accuracy on ImageNet-C under label shifts using ResNet.
> |Method|Time (s)|Accuracy (%)|
> |-|:-:|:-:|
> |Tent|110|22.8|
> |ETA|112|26.2|
> |EATA|118|31.7|
> |ReCAP|116|47.2|
> >Q4: It is important, given the efficiency of ReCAP, to show the performance gain under computational budgeted evaluation.
>
> A4: Thank you for your valuable suggestion. We agree that this evaluation is essential and realistic for assessing TTA methods. As shown in Tab. 2, ReCAP benefits from the computational efficiency of the upper-bound proxy, resulting in minimal performance degradation while achieving more significant gains under strict time constraints. This demonstrates ReCAP's ability to provide efficient adaptation under time limitations, making it well-suited for real-world deployments with computational budgets.
>
> Table 2: Error rate on ImageNet-C under computational time constraints.
> |Method|Realistic|Gaus.|Shot|Impu.|Defo.|Glas.|Moti.|Zoom|Snow|Fros.|Fog|Brig.|Cont.|Elas.|Pixe.|Jpeg|Avg.|
> |-|:-:|-|-|-|-|-|-|-|-|-|-|-|-|-|-|-|-|
> |EATA|✗|65.5|62.4|63.5|66.6|67.2|52.0|47.3|48.2|54.1|39.9|32.1|55.0|42.3|39.2|44.8|52.0|
> |EATA|✓|69.3|67.1|69.2|71.1|71.7|57.5|49.9|51.9|57.4|42.4|32.6|60.7|45.1|41.4|47.4|55.6(+3.6)|
> |SAR|✗|69.5|69.7|69.0|71.2|71.7|58.1|50.5|52.9|57.9|42.7|32.7|62.9|45.5|41.6|47.8|56.2|
> |SAR|✓|79.4|78.5|78.1|79.9|79.3|67.5|56.1|60.5|63.1|47.4|34.0|75.3|51.7|46.6|53.8|63.4(+7.2)|
> |DeYO|✗|64.1|61.4|62.1|66.0|66.2|51.7|47.4|47.5|54.0|39.8|31.9|54.0|41.9|38.7|44.3|51.4|
> |DeYO|✓|69.7|67.6|68.2|73.2|72.2|59.0|50.8|52.8|58.1|42.7|32.5|62.9|45.5|41.5|48.1|56.3(+4.9)|
> |Ours|✗|64.1|60.4|62.1|67.0|67.2|50.6|47.2|45.8|51.7|38.2|32.2|53.5|41.8|38.4|43.9|50.9|
> |Ours|✓|68.2|65.2|67.1|70.7|71.0|55.7|49.8|50.0|53.8|40.6|32.7|52.9|44.9|40.6|46.7|54.0(+3.1)|
> >Q5: One extra [Optional] experiment is to extend the evaluation to the Practical TTA setting which is closely related to the wild TTA setting.
>
> A5: Thank you for this insightful suggestion. We agree that the PTTA setting (Continual + Label Shifts) is closely related to the wild TTA setting. As shown in Tab. 3, despite ReCAP not incorporating any additional design specifically for continual adaptation, it still outperforms entropy-based methods and specific-design RoTTA in PTTA setup. This result further validates the effectiveness and robustness of ReCAP across diverse test-time conditions.
>
> Table 3: Accuracy on ImageNet-C under the PTTA setup, evaluated on ResNet50.
> |Method|Gaus.|Shot|Impu.|Defo.|Glas.|Moti.|Zoom|Snow|Fros.|Fog|Brig.|Cont.|Elas.|Pixe.|Jpeg|Avg.|
> |-|:-:|:-:|:-:|:-:|:-:|:-:|:-:|:-:|:-:|:-:|:-:|:-:|:-:|:-:|:-:|:-:|
> |Source|17.9|19.9|17.9|19.7|11.3|21.3|24.9|40.4|47.4|33.6|69.2|36.3|18.7|28.4|52.2|30.6|
> |Tent|13.7|0.9|0.2|3.0|0.4|0.3|0.4|0.6|0.2|0.2|1.7|0.4|0.1|0.2|1.2|1.6|
> |SAR|32.0|14.0|17.7|16.7|12.6|1.1|16.5|44.5|42.4|11.1|7.7|46.6|8.6|0.6|38.6|20.7|
> |DeYO|40.7|44.1|41.3|17.7|22.1|41.3|16.5|41.2|50.5|30.9|73.2|51.4|42.4|56.5|58.2|41.9|
> |RoTTA|40.2|41.2|40.8|20.7|20.3|40.2|33.2|45.2|51.1|52.1|70.2|50.1|40.1|52.1|57.1|43.6|
> |ReCAP|42.2|44.3|42.1|18.9|23.9|42.1|28.7|44.7|51.6|52.5|71.2|52.2|41.5|57.9|58.3|44.8|

---

> > ### Comment · Reviewer_eZDc · 2025-04-07
> >
> > I would like to thank the authors for their efforts in replying to my comments. My questions were adequately answered. Thus, I am raising my score from weak accept to Accept.

---

> > > ### Author Response · Authors · 2025-04-07
> > >
> > > We are glad to know that our response has addressed your questions.
> > >
> > > We sincerely appreciate your insightful and constructive feedback. Your comments have guided us to refine our work and better articulate the significance of our contributions.
> > >
> > > Once again, thank you for your thoughtful evaluation and recognition of our work!
> > >
> > > Best regards,
> > >
> > > The Authors

---

### Official Review · Reviewer_7x3y · 2025-03-12

**Overall Recommendation:** 4

**Summary:**

This paper proposes a new method, ReCAP, a novel approach to addressing the main limitation of TTA in entropy minimization. The key idea of this work is that EM heavily relies on local consistency, and when this consistency is disrupted, model performance degrades. To resolve this issue, instead of optimizing the confidence of individual samples, ReCAP optimizes region-based confidence using bias and variance terms through Region Confidence Optimization. Furthermore, to enable low-cost computation and accuracy, the method employs approximation theories (e.g., Finite-to-Infinite Approximation). When applied to low-data settings (batch size = 1), the proposed method demonstrated a +3.5% improvement in performance.

**Claims And Evidence:**

The paper experimentally demonstrates that entropy minimization leads to performance degradation when local consistency is disrupted. The results show that even when entropy values are similar, prediction differences can be significant in domain-shift environments. Furthermore, the proposed RCO method improves the stability of TTA, and ReCAP outperforms traditional entropy-based methods such as Tent and MEMO, proving to be particularly effective in domain shift scenarios. The study also validates that the Bias Term and Variance Term play a crucial role in maintaining prediction consistency through mathematical formulations and empirical analysis. Additionally, the paper demonstrates that ReCAP achieves higher performance than Tent with only a 5% increase in computational cost.

**Essential References Not Discussed:**

This paper leverages the appendix to cite all relevant studies comprehensively.

**Experimental Designs Or Analyses:**

The experimental design appears relatively reliable, demonstrating that ReCAP maintains high performance even in low-data settings and remains stable across various corruption types and domain shift scenarios. The study also presents t-SNE visualizations, confirming that ReCAP enhances class separability. Overall, the experiments are appropriately designed to support the paper’s claims.

**Methods And Evaluation Criteria:**

The study employs widely used datasets in TTA research, including ImageNet-C, ImageNet-R, and VisDA-2021, to evaluate performance. Comparisons are made with state-of-the-art TTA techniques such as Tent, MEMO, DDA, EATA, SAR, and DeYO. The evaluation focuses on improvements in accuracy under domain shifts, robustness in low-data scenarios, and performance across mixed-domain tests. The inclusion of an ablation study analyzing key hyperparameters such as region size and bias-variance tradeoff strengthens the validity of the evaluation framework, aligning well with the study’s research objectives.

**Other Comments Or Suggestions:**

- It would be helpful if the paper clarified what value of 𝜏  was fixed when conducting experiments on the effect of 𝜆 in Section 6.1.
- The t-SNE plots effectively illustrate the improvements in feature space adaptation, making it easier to understand the impact of ReCAP on prediction consistency and clustering quality.

**Other Strengths And Weaknesses:**

Strength
- While entropy minimization has been used in TTA, its accuracy gains have been limited. This paper provides a meaningful finding by identifying its limitations and proposing an effective solution.
- The paper effectively explains why local consistency is critical in TTA and thoroughly discusses the limitations of existing methods, making a strong case for the necessity of ReCAP.
- Instead of optimizing confidence at the sample level, the paper introduces region-based confidence optimization, which is a more robust and reliable strategy for TTA.
- The paper rigorously evaluates ReCAP across various datasets and settings, including different domain shifts, data scarcity scenarios, and mixed-domain testing. This strengthens the credibility of the proposed method and demonstrates its robustness in real-world applications.
- Unlike computationally intensive methods like DDA, ReCAP maintains a lightweight adaptation process while still improving accuracy.
- ReCAP not only outperforms baseline methods but also enhances other approaches such as SAR and DeYO, demonstrating its adaptability and versatility

Weakness
- While the method is effective for classification, it is unclear how well it would generalize to more complex tasks like object detection, segmentation, or NLP.
- The paper assumes that the finite-to-infinite approximation holds consistently, but in scenarios where domain shifts occur rapidly, there is a possibility that this assumption might not always hold. Investigating its robustness in highly dynamic environments could provide further insights.
- While the paper discusses applying features to reduce computational cost, providing a quantitative comparison of the actual reduction in computation would strengthen the analysis.

**Questions For Authors:**

- Does the variance + bias function serve the same role as the traditionally used mutual information?
- How does ReCAP handle rapid domain shifts where local consistency may be entirely lost?
- Could ReCAP be extended to structured prediction tasks like segmentation?
- How sensitive is ReCAP to the choice of when adapting to new domains?

**Relation To Broader Scientific Literature:**

This work builds upon prior entropy minimization-based TTA research, such as Tent, MEMO, and EATA, extending the optimization approach from sample-level to region-level confidence estimation. Additionally, it is relevant to domain adaptation research, distinguishing itself by focusing on maintaining local consistency as a key factor in adaptation performance.

**Theoretical Claims:**

The paper theoretically supports its approach by introducing an optimization framework leveraging Bias and Variance Terms to balance confidence estimation and prediction consistency. Additionally, it proposes a Finite-to-Infinite Approximation method to reduce computational cost while effectively approximating regional confidence. The mathematical derivations appear valid, and the experimental results substantiate the proposed theoretical foundation.

---

> ### Author Rebuttal · Authors · 2025-03-31
>
> We appreciate your detailed review and positive feedback on our contributions, including meaningful findings, novel region-based confidence optimization, and comprehensive evaluation. Building on your comments, we provide additional explanations and experiments to further demonstrate ReCAP's effectiveness and efficiency.
> >Q1: Generalization ability in more complex tasks.
>
> A1: Thank you for raising this important point. While our current experiments focus on classification, the core idea of region-based confidence optimization is inherently versatile. Additional evaluations on segmentation (Tab. 1 in response to Reviewer ynrW) and object detection (Tab. 1 in this response) show consistent improvements of ReCAP over entropy-based methods, indicating that ReCAP can be integrated into diverse model architectures and effectively extended to various complex tasks.
>
> Table 1: Comparisons of detection performance on KITTI-C benchmark in [1] with MonoFlex, regarding AP.
> |Method|Gauss.|Shot|Impul.|Defoc.|Glass|Motion|Snow|Frost|Fog|Brit.|Contr.|Pixel|Sat.|Avg|
> |-|:-:|:-:|:-:|:-:|:-:|:-:|:-:|:-:|:-:|:-:|:-:|:-:|:-:|:-:|
> |Source|4.2|7.5|5.6|2.6|3.8|10.9|15.6|10.5|7.5|24.8|7.1|29.1|31.9|12.4|
> |TENT|16.0|25.1|23.8|21.7|11.6|27.1|26.9|26.9|30.5|35.8|33.7|41.1|35.2|27.3|
> |EATA|16.8|25.9|24.7|22.1|13.6|27.5|27.7|27.4|30.7|35.6|33.9|41.0|35.6|27.9|
> |DeYO|19.2|26.1|24.7|23.2|15.6|28.5|28.5|29.3|30.8|35.1|34.2|40.8|36.2|28.6|
> |MonoTTA (latest SOTA)|21.3|28.2|26.2|25.8|19.4|31.8|29.3|30.2|32.1|36.1|36.5|41.2|37.4|30.4|
> |ReCAP|21.3|29.3|26.3|26.7|20.1|31.1|32.2|32.6|31.7|36.7|36.1|41.3|37.5|31.0|
> >Q2: How does ReCAP handle highly dynamic shifts where local consistency may be entirely lost?
>
> A2: Thank you for your insightful question. Our finite-to-infinite approximation is derived without assuming any consistency condition, ensuring its applicability even when consistency is entirely lost. Based on this foundation, ReCAP employs region-confidence optimization to enhance local consistency, which is crucial for robust adaptation.
>
> Moreover, we evaluate ReCAP in a highly dynamic setting where the data stream undergoes rapid transitions across different domains, including style, corruption, and label shifts (see Appendix B.1). The results demonstrate that ReCAP exhibits strong robustness and achieves SOTA performance, validating its capability to address highly dynamic scenes.
> >Q3: Quantitative comparison of computational cost reduction.
>
> A: ReCAP reduces the computational cost via feature-level region modeling, eliminating the overhead of image-level region modeling and augmentation. Furthermore, its finite-to-infinite approximation serves as an efficient proxy, removing the need for costly sampling. As shown in Tab. 2, these designs achieve significant runtime reduction. We will incorporate this quantitative comparison into the revised version to enhance clarity on the efficiency of ReCAP.
>
> Table 2: Running time on 50,000 images.
> |Region Type|Time (s)|
> |-|:-:|
> |Image-level region (16 augmentation)|1798|
> |Feature-level region (w/o proxy, 16 sampling)|163|
> |Feature-level region (w/ proxy)|116|
> >Q4: Clarification on ablation study settings.
>
> A4: In our analysis of the effect of λ in Section 6.1, we fixed τ at 1.2, which aligns with the default value used across all experiments. We will explicitly state it in the revised version.
> >Q5: Variance + Bias vs. Mutual Information.
>
> A5: Our variance + bias function serves a fundamentally different role from mutual information (MI) in several key aspects:
> 1. Different Objects: MI is defined between two random variables, measuring their shared information, whereas our variance + bias function is computed over a local region surrounding a single sample x, capturing localized prediction stability.
> 2. Different Purposes: MI primarily quantifies mutual dependence, while our function serves as prediction confidence and consistency measure within a local feature region, making it more aligned with adaptation objectives.
> 3. Different Optimization Effects: MI encourages statistical association but does not address prediction probability discrepancies. In contrast, our function directly optimizes both prediction uncertainty (bias) and local discrepancy (variance), enhancing robustness under domain shifts.
>
> >Q6: Sensitivity of RaCAP.
>
> A6: We have extensively evaluated ReCAP across diverse datasets (ImageNet-C, VisDA, ImageNet-R), tasks (classification, segmentation, detection), TTA settings (wild, mild, and continual), and hyperparameter configurations. Our results consistently show its robustness and reliability across these various conditions.
>
> Additionally, we notice that the question might miss a word (e.g., choice of L_0 when adapting). If we have misunderstood your concern, please clarify, and we would be happy to provide further insights.
> >References
> [1] Lin, Hongbin, et al. "Monotta: Fully test-time adaptation for monocular object detection." European Conference on Computer Vision, 2024.

---

> > ### Comment · Reviewer_7x3y · 2025-04-04
> >
> > Thank you for the response—especially for including additional experimental results and for the detailed explanation on how your method differs from mutual information. The clarification regarding the finite-to-infinite approximation also helped me better understand your formulation.
> >
> > Also, to follow up on my final question (Q6), I realized that I had originally meant to refer to the τ(tau), which I mistakenly left out—apologies for the confusion. The ablation study (Fig. 4b) shows stable performance within a reasonable τ range, supporting the method’s robustness. However, the performance drop beyond τ = 2.5 raises questions, how sensitive the method is in real-world scenarios where the optimal τ may not be known in advance. It would be helpful to better understand how much τ influences performance in practice.
> >
> >
> > ---------
> > Update following "Reply Rebuttal Comment by Authors":
> >
> > Thank you for your thoughtful responses to my final questions. The additional experiments across diverse domains consistently show strong performance around similar τ values were convincing. I recognize the strength of your work and have decided to raise my score to a 4.

---

> > > ### Author Response · Authors · 2025-04-05
> > >
> > > Thank you very much for your constructive and positive feedback on our response. Following your suggestions, our additional experiments and explanations have further strengthened this work, particularly in terms of its broader applicability and high efficiency. We are also grateful for your clarification on Q6 and hope to address your question below:
> > >
> > > >Q7:  The ablation study (Fig. 4b) shows stable performance within a reasonable τ range, supporting the method's robustness. However, the performance drop beyond τ = 2.5 raises questions, how sensitive the method is in real-world scenarios where the optimal τ may not be known in advance.
> > >
> > > A7: Thank you for recognizing the robustness of our method. To further clarify the practical stability of τ selection, we provide additional discussion and a practical example below:
> > >
> > > 1. **Default Value as a Reliable Choice:**  Across all experiments in our manuscript, we consistently use a fixed τ=1.2, which delivers SOTA results across various datasets and TTA scenarios. This value serves as a reliable choice, and we recommend its use in cases where a validation set is unavailable.
> > >
> > > 2. **Stable Optimal Range:** Hyperparameter tuning on a small validation set (10% Gaussian-type data from ImageNet-C) across 3 settings and 2 model architectures consistently selects 1.2 ($\pm 0.2$). Further validation reveals that values within $\[0.6, 1.6\]$ maintain strong performance (Tab. 3), confirming a stable optimal region for reliable adaptation.
> > >
> > > 3. **Real-World Practicality:** To assess τ sensitivity in real-world scenarios,  we examine its impact on a detection task. The default τ=1.2 achieves 31.0 AP on KITTI-C, outperforming prior SOTA (Tab. 1). Additionally, grid search over $\[0.6, 1.6\]$ on the validation set selects τ=1.3, improving AP to 31.2. This supports our default setting as a strong baseline and our tuning range as a practical search space.
> > >
> > > Thank you again for raising this critical point. Given the empirical evidence and actionable guidance provided, we believe that τ selection is both stable and practical for researchers and practitioners, ensuring reliable performance without excessive sensitivity concerns.
> > >
> > > Table 3: Additional ablation study on τ under 3 settings and 2 models. Results that surpass the prior SOTA are in **bold**.
> > > |Setting|τ=0.6|τ=0.8|τ=1.0|τ=1.2|τ=1.4|τ=1.6|
> > > |-|:-:|:-:|:-:|:-:|:-:|:-:|
> > > |Batch Size=1 (ResNet)|**46.6**|**47.3**|**47.5**|**47.6**|**47.2**|**46.3**|
> > > |Batch Size=1 (ViT-Base)|**64.2**|**64.9**|**65.4**|**65.6**|**65.5**|**65.1**|
> > > |Mixed Domain=1 (ResNet)|**40.0**|**41.2**|**42.0**|**42.1**|**42.1**|**42.0**|
> > > |Mixed Domain=1 (ViT-Base)|**58.6**|**59.5**|**59.5**|**59.6**|**59.5**|**59.5**|
> > > |Label Shift=1 (ResNet)|**45.5**|**46.6**|**47.1**|**47.2**|**46.4**|**45.3**|
> > > |Label Shift=1 (ViT-Base)|**61.5**|**62.1**|**62.6**|**63.0**|**62.6**|**62.2**|
> > >
> > > Again, we would like to thank you for appreciating our work and recognizing our contributions!
> > >
> > > Best regards, Authors

---

### Official Review · Reviewer_6Q1i · 2025-03-14

**Overall Recommendation:** 3

**Summary:**

This paper proposes a region modification based mechanism, called “Region Confidence Adaptive Proxy (ReCAP), to address the problem of will test-time adaptation (WTTA). Further, it develops a finite-to-infinite asymptotic approximation, which is a tractable upper bound to the intractable region confidence.
Experimental results show improved performance of ReCAP compared to other approaches.

**Claims And Evidence:**

Yes.

**Essential References Not Discussed:**

NA

**Experimental Designs Or Analyses:**

The experiments follow the existing WTTA line of work.

**Methods And Evaluation Criteria:**

Yes.

**Other Comments Or Suggestions:**

1. Line 235-236: It should be Eq. 5 in place of Eq. 10.
2. The recent focus in the area of test-time adaptation has shifted towards continual test-time adaptation (CTTA), so experiments in the CTTA setting will enhance the contribution of this paper.

**Other Strengths And Weaknesses:**

**Strengths**
* Proposed a tractable upper bound to the intractable region confidence.
* The theoretical results are interesting.
* Most of the experimental results show improvements.

**Weaknesses**
* Some of the empirical gains are marginal; for example, in Table 2, VitBase, DeYO -> ReCAP gain is =< 0.5.
* Limited real-world applicability of WTTA, with batch size = 1 setting.

**Questions For Authors:**

1. How is the hyperparameter L_0 in equation 9 tuned for the experiments?
2. Is the accuracy in Figure 4 measured on the test set itself, on which the final performance is reported? Whether there is a validation split for tuning?
3. In real-world applications, isn't the setting, such as batch size = 1, too contrived, though it is challenging? Can we not accumulate more examples before updating, effectively increasing the batch size?

**Relation To Broader Scientific Literature:**

The problem of WTTA is challenging. However, with the advent of more realistic continual test-time adaptation [1] approaches, the real world applicability of WTTA seems limited compared to recent progress.

References:
1. Wang, Qin, et al. "Continual test-time domain adaptation." Proceedings of the IEEE/CVF Conference on Computer Vision and Pattern Recognition. 2022.

**Theoretical Claims:**

The theoretical claims in the main paper have been checked. The details of the proofs in Appendix have not been thoroughly verified.

---

> ### Author Rebuttal · Authors · 2025-03-31
>
> Thank you for carefully reviewing our paper and offering a positive assessment. We appreciate your recognition of the contributions made by our work, particularly the idea of the tractable bound on the intractable region confidence and the theoretical results.
> >Q1: Some of the Empirical Gains are marginal. For example, VitBase in mixed testing domain.
>
> A1: Thank you for your feedback. In highly competitive TTA scenarios, performance gains tend to approach saturation in some cases. However, larger domain gaps, such as more severe corruption or mixed style shifts, still present significant challenges in terms of adaptation efficiency and robustness for TTA methods.
>
> To access the empirical gains under more severe shifts, we increase the severity level from 4 & 5 to levels 6 & 7 (see Tab. 1), and our method achieves significant improvements of **+5.7** and **+5.2** over DeYO. Furthermore, under complex style shifts, ReCAP achieves average gains of **+2.6** on ImageNet-R and **+1.7** on VisDA (Appendix B.1). Overall, our method consistently outperforms prior methods across 3 datasets, 3 wild settings, and 2 base models, achieving gains of >+1.5 in the majority of scenarios.
>
> Table 1: Comparisons on ImageNet-C (severity level 6, 7) using VitBase under Mixed Testing Domain.
> |Method|Level 6|Level 7|
> |-|-|-|
> |Source|18.87|12.80|
> |TENT|2.45|0.99|
> |EATA|30.58|16.32|
> |SAR|32.11|17.74|
> |DeYO|29.54|16.08|
> |ReCAP|**35.27**|**21.32**|
> >Q2: Limited applicability of bs=1 setting. Can we not accumulate more examples to increase the batch size?
>
> A2: We appreciate your comment. While bs=1 may seem contrived, some real-world applications (e.g., edge computing) face hardware constraints that necessitate the use of small mini-batches. Following your comment, we evaluate the effect of accumulating examples with varying sizes (See Tab. 2). However, small batch sizes still present a crucial bottleneck, hindering adaptation performance. This underscores the importance of developing robust TTA solutions tailored to such restrictive conditions.
>
> Table 2: Accuracy of Tent on ImageNet-C across different accumulated batch sizes, evaluated on ResNet50.
> ||no-adapt|bs=1|bs=4|bs=16|bs=32|bs=64|
> |:-:|:-:|:-:|:-:|:-:|:-:|:-:|
> |Tent (%)|30.6|21.5|23.5|25.9|28.6|33.9|
> >Q3: Line 235-236: It should be Eq. 5 in place of Eq. 10.
>
> A3: Thank you for pointing out this typo, and we will correct it.
> >Q4: Additional experiments in the CTTA setting will enhance the contribution of this paper.
>
> A4: Thank you for your constructive suggestion. Following your advice, we evaluate our method in CTTA scenarios for classification (Tab. 3 in this response) and semantic segmentation (Tab. 1 in response to Reviewer ynrW). While ReCAP is not designed for CTTA setup, it shows competitive performance and outperforms several strong baselines. These additional evaluations further highlight the broad applicability of our method across mild, continual, and wild settings.
>
> Table 3: Error rate (%) in CTTA scenario (CIFAR100C) [1], evaluated on ResNeXt-29.
> |Method|Gaus.|Shot|Impu.|Defo.|Glas.|Moti.|Zoom|Snow|Fros.|Fog|Brig.|Cont.|Elas.|Pixe.|Jpeg|Avg|
> |-|-|-|-|-|-|-|-|-|-|-|-|-|-|-|-|-|
> |Source |73.0|68.0|39.4|29.3|54.1|30.8|28.8|39.5|45.8|50.3|29.5|55.1|37.2|74.7|41.2|46.4|
> |TENT|37.2|35.8|41.7|37.9|51.2|48.3|48.5|58.4|63.7|71.1|70.4|82.3|88.0|88.5|90.4|60.9|
> |CoTTA|40.1|37.7|39.7|26.9|38.0|27.9|26.4|32.8|31.8|40.3|24.7|26.9|32.5|28.3|33.5|32.5|
> |SAR|39.7|34.3|36.5|26.4|37.4|28.6|26.1|32.7|31.4|36.6|26.1|29.6| 33.0|29.8|38.1|32.4|
> |EcoTTA|39.1|35.7|37.5|26.2|37.7|28.3|26.3|32.2|31.0|36.9|25.9|27.4|32.7|28.4|34.7|32.0|
> |DeYO|39.0|34.1|36.3|26.7|37.2|28.4|26.2|32.4|31.6|36.2|25.5|26.8|32.2|30.1|38.3|32.1|
> |ReCAP|38.8|33.5|36.5|26.5|37.9|28.2|26.4|31.1|29.6|34.0|25.8|27.7|32.0|28.2|38.1|31.4|
> >Q5: How is the hyperparameter L_0 in equation 9 tuned for the experiments?
>
> A5: For hyperparameter L_0, we perform a grid search over a range of values on a small validation set, which comprises 10% of the Gaussian-type data from ImageNet-C. Additionally, we conduct a sensitivity analysis (see Appendix C.1) to confirm the robustness of our method to variations in L_0. Further details on this tuning process will be included in the revised version.
> >Q6: Is the accuracy in Figure 4 measured on the test set itself, on which the final performance is reported? Whether there is a validation split for tuning?
>
> A6: For the sensitivity analysis in Figure 4, we measure accuracy on the entire test set to validate the robustness of our method. For hyperparameter tuning, we use a small validation split (the same set used for L_0). The hyperparameters selected through this process are then validated and shown to be robust in Figure 4. We will provide a clearer explanation of this procedure in the revised version to avoid any confusion.
> >References
> [1] Wang, Qin, et al. "Continual test-time domain adaptation." Proceedings of the IEEE/CVF Conference on Computer Vision and Pattern Recognition. 2022.

---

> > ### Comment · Reviewer_6Q1i · 2025-04-07
> >
> > Thanks to the authors for their response.
> >
> > I do not have any further queries or comments.

---

> > > ### Author Response · Authors · 2025-04-08
> > >
> > > We sincerely express our gratitude for your valuable feedback. Thanks to additional discussions and experiments, we were able to effectively convey the contributions of our work.
> > >
> > > Again, we would like to thank you for appreciating our work and recognizing our contributions!
> > >
> > > Best,
> > >
> > > The Authors

---

### Decision · Program_Chairs · 2025-05-01

**Decision:**

Accept (poster)

**Comment:**

Test-time adaptation aims to reduce generalization error by updating on shifted data. This work analyzes the "mild" (= single shift) and wild (= multiple shift, and small batches) settings for test-time adaptation and finds that local inconsistency is a challenge. The proposed method, ReCap (Region Confidence with Adaptive Proxy), identifies more local regions in which to optimize instead of simply minimizing entropy or some thresholding of entropy as in prior work. Experiments show improvement on the common benchmark of ImageNet-C, in comparable settings to prior work including with batch size one and with imbalanced classes, and with comparison to recent and strong methods. Furthmore ReCAP can be combined with base methods, like SAR and DeYO, and shows further improvement. This is a clear accept given the contributions and the reviewer consensus for acceptance (4/4 agree with scores of 4, 4, 4, 3). The authors provided a rebuttal, all reviewers acknowledged it, and eZDc + 7x3y + ynrW raise 3 to 4 and 6Q1i maintains 3, so the consensus is clear, and the area chair agrees with acceptance.